# ORACLE EFFICIENT TRUNCATED STATISTICS

**Konstantinos Karatapanis**[1,2]**, Vasilis Kontonis**[3]**, Christos Tzamos**[1,4]
[1] Archimedes, Athena Research Center, Greece   [2] National Technical University of Athens
[3] University of Texas at Austin   [4] University of Athens
k.karatapanis@athenarc.gr, vasilis@cs.utexas.edu, chtzamos@di.uoa.gr

## ABSTRACT

We study the problem of learning from truncated samples: instead of observing samples from some underlying population $p^*$, we observe only the examples that fall in some survival set $S \subset \mathbb{R}^d$ whose probability mass (measured with respect to $p^*$) is at least $\alpha$. Assuming membership oracle access to the truncation set $S$, prior works obtained algorithms for the case where $p^*$ is Gaussian Daskalakis et al. (2018) or more generally an exponential family with strongly convex likelihood Lee et al. (2023) — albeit with a super-polynomial dependency on the (inverse) survival mass $1/\alpha$ both in terms of runtime and in number of oracle calls to the set $S$. In this work we design a new learning method with runtime and query complexity polynomial in $1/\alpha$. Our result significantly improves over the prior works by focusing on efficiently solving the underlying optimization problem using a general purpose optimization algorithm with minimal assumptions.

## 1 INTRODUCTION

We study the problem of inference from truncated samples: assuming an underlying population $p$ over data in $\mathbb{R}^d$ the learner has access to samples from the conditional measure $p^S$ over some survival set $S \subseteq \mathbb{R}^d$. This conditioning or truncation of the underlying population data may be the outcome of censorship, imperfect data collection processes, measurement errors, user preferences, etc. Inference from truncated data is a central problem in statistics going back to fundamental works in the beginning of the previous century Galton (1898); Pearson (1902b); Pearson & Lee (1908b). Due to its practical importance, more recently, the problem of inference from truncated data has seen significant developments with a focus on providing *computationally efficient estimation algorithms* Daskalakis et al. (2018); Kontonis et al. (2019); Ilyas et al. (2020); Daskalakis et al. (2020b;a; 2021); Lee et al. (2023).

In this work we consider the problem of fitting a parametric model to truncated data given oracle access to the survival set $S$ (i.e., the set of examples that are *not truncated*). In this model Daskalakis et al. (2018) was the first work that gave a computationally and statistically efficient algorithm for learning truncated Gaussian distributions with membership oracle access to the truncation set. More recently, the work Lee et al. (2023) extended those results and gave efficient algorithms when the underlying population follows an exponential family. In this work we study learning truncated exponential families with computationally and statistically efficient algorithms that perform well even when the fraction of observed data is very small (compared to the full population).

**Learning truncated exponential families** An exponential family $p_\theta$, parameterized by some $k$-dimensional vector $\theta \in \Theta$ is a density of the form $p_\theta(x) = h(x) \exp(\theta^\top T(x) - A(\theta))$, where $h(x) : \mathbb{R}^d \mapsto [0, \infty)$ is a weight function, $T(x) : \mathbb{R}^d \mapsto \mathbb{R}^k$ is the sufficient statistic, $A(\theta) = \log \left( \int h(x) \exp(\theta^\top T(x)) dx \right)$ is the appropriate normalization constant (aka the log-partition function). Given an exponential family $p_\theta$ and a survival set $S$, we define the corresponding truncated exponential family as $p_\theta^S(x) = p_\theta(x) \mathbb{1}_S(x) / \int_S p_\theta(x) dx$, where $\mathbb{1}_S(x)$ denotes the indicator function of whether $x$ belongs in $S$. Given samples from some truncated exponential family $p_{\theta^*}^S$ the goal of the learner is to identify an approximation to the (unknown) parameter $\theta^*$. Given some target accuracy $\epsilon > 0$, the approximation guarantees for the learned $\theta$ can either be in some distance over the parameter space, i.e., $\|\theta - \theta_*\| \leq \epsilon$ (parameter recovery) or over distributions, e.g., that the total variation distance between $p_\theta$ and $p_{\theta^*}$ is at most $\epsilon$. The goal is to find a good candidate parameter $\theta$ with as few (i) samples from $p_{\theta^*}^S$; (ii) oracle calls to $\mathbb{1}_S(x)$ and (iii) computational resources as possible.

The fraction of observed data or equivalently the probability that a sample from the underlying population $p_{\theta*}$ is truncated, captures the difficulty (both statistical and computational) of inference from truncated data. More precisely, we define $\alpha = p_{\theta*}(S)$ to be the probability assigned to the survival set by the unknown exponential family $p_{\theta*}$. While previous works Daskalakis et al. (2018); Lee et al. (2023) on learning truncated exponential families have polynomial dependencies on the parameter dimension $k$ and the accuracy parameter $\epsilon$, their dependency on the survival probability $\alpha$ is far from optimal. In particular, their runtime, sample complexity, and oracle calls all depend super-polynomially (i.e., $(1/\alpha)^{\text{poly}(\log(1/\alpha))}$) to the survival probability $\alpha$. In this work our focus will be to provide algorithms that achieve dependencies on the dimension $k$ and accuracy $\epsilon$ but also on the survival mass $\alpha$.

> *Is there a computationally efficient algorithm that can learn truncated exponential families with a polynomial dependence on the survival mass $1/\alpha$? Can we get state of art dependencies on dimension and accuracy parameters at the same time?*

Aiming for algorithms with low sample-complexity is perhaps the most well-studied task from a statistical perspective and is of great importance for virtually every inference task. We remark that in the context of truncated statistics getting oracle efficient algorithms is also important as the membership oracle may correspond to a costly or complex mechanism (e.g., we may need to find candidates to complete a new questionaire, perform a new physical experiment, etc.).

Our main contribution is a positive answer to the above question by providing a new analysis of the truncated negative log likelihood objective.

## 1.1 OUR CONTRIBUTIONS

We first formally define the class of exponential family distributions that we consider in our work. Similarly to Lee et al. (2023), we assume that the non-truncated negative log-likelihood of the exponential family is strongly convex and smooth as a function of the parameter $\theta$. Given some exponential family $p_\theta$ parameterized by $\theta \in \mathbb{R}^k$, the negative log-likelihood (NLL) over a population distribution $q$ is defined as follows:

$$\mathcal{L}(\theta) = - \mathop{\mathbf{E}}_{x \sim q}[\log p_\theta(x)]. \tag{1}$$

In what follows we will refer to $\mathcal{L}$ as "non-truncated". Moreover, for a truncated exponential family $p_\theta^S$ we define the following "truncated" version of the NLL objective as:

$$\mathcal{L}_S(\theta) = - \mathop{\mathbf{E}}_{\mathbf{x} \sim q}[\log p_\theta^S(x)] = \mathcal{L}(\theta) + \log p_\theta(S). \tag{2}$$

**Definition 1.1** (Strongly convex and smooth exponential families). *We assume that $\mathcal{L}(\theta)$ is $\lambda$-strongly convex and $L$-smooth as a function of $\theta$, i.e.,*

$$\lambda I \preceq \mathbf{Cov}_{x \sim p_\theta}[T(x), T(x)] \preceq LI.$$

**Remark 1.2.** *Lee et al. (2023) further places strong structural assumptions on $p_\theta$. In particular, that the underlying non-truncated exponential family is log-concave (i.e., its log-density is concave as a function of $x$) and also that the sufficient statistic $T(x)$ is a polynomial function of $x$. As we will see our main learning result is quite general by providing a weaker learning guarantee (than parameter recovery) and therefore requires no such assumptions a priori. However, given those assumptions we can obtain similar guarantees to the prior works Daskalakis et al. (2018); Lee et al. (2023).*

We remark that, as observed in Lee et al. (2023) the above assumption is quite general and captures many well-studied exponential families such as Gaussians, Exponentials, Weibull, (Continuous) Poisson, etc.

Since our focus in this work is learning (as opposed to sampling from) exponential families, we treat the sample generation process as a black-box and define the following sampling oracle that returns samples from a given exponential family or truncated exponential family. For the truncated case, we crucially require that the learner must give the oracle some $p_\theta$ that assigns non-trivial mass to the survival set $S$. This is to ensure that our sampling oracle can be readily implemented via rejection sampling without a large number of rejected samples (that would in turn require a large number of calls to the membership oracle to $S$). For example, assume that the survival set is $S = [0, 1]$ and

the ground-truth exponential family is a normal distribution (not necessarily with unit variance) that assigns $\alpha$ mass to the set $S$. The learner can request samples from unit variance normal distributions whose means are roughly within $O\left(\sqrt{\log(1/\alpha)}\right)$ distance from 0 but not, for example, from a normal centered at $-1/\alpha$ which puts exponentially small (i.e.,$2^{-\Omega(1/\alpha^2)}$) mass to $S$.

**Definition 1.3** (Sample Oracles). *Given as input an exponential family $p_\theta \in \mathcal{F}$, the (non-truncated) sample oracle returns a sample $x \sim p_\theta$.*

*Moreover, given as input an exponential family $p_\theta \in \mathcal{F}$, the $\beta$-truncated sample oracle over a survival set $S$ returns a sample $\mathbf{x} \sim p_\theta^S$, provided that $p_\theta(S) \geq \beta$. In case this constraint does not hold the behavior of the oracle is undefined.*

**Remark 1.4** (Implementing a $\beta$-Truncated Sample Oracle). *Given a membership oracle to the set $S$ and a sampler for the non-truncated exponential family $p_\theta$, one can directly implement a $\beta$-truncated sample oracle by rejecting samples until one falls in $S$.*

We now present our main result on learning truncated exponential families that requires roughly $\frac{k}{\epsilon^2}\text{poly}(\log(1/\alpha))$ samples from the unknown truncated density $p_{\theta^*}^S$ and makes $\frac{k}{\epsilon^2}\text{poly}(\log(1/\alpha))$ sample-oracle calls to an $\alpha$-truncated sample oracle of Definition 1.3 to learn a truncated density close to the observed truncated distribution in Kullback-Leibler (KL) divergence.

**Theorem 1.5** (Truncated Exponential Families). *Suppose $F$ is an exponential family whose negative log-likelihood (NLL), $\mathcal{L}(\theta)$, satisfies the conditions in Definition 1.1. Then, there exists an algorithm that uses $\widetilde{O}\left(\frac{k}{\epsilon^2}\log\left(\frac{1}{\alpha}\right)\right)$ samples from $p_{\theta^*}^S$, makes $\widetilde{O}\left(\frac{k}{\epsilon^2}\log\left(\frac{1}{\alpha}\right)^2\right)$ calls to an $\Omega(\alpha)$ sample oracle (of Definition 1.3) runs in time $\text{poly}(k, 1/\epsilon, \log(\frac{1}{\alpha}))$ and computes an estimate $\hat{\theta}$, such that $\text{KL}(p_{\theta^*}^S\|p_{\hat{\theta}}^S) \leq \epsilon$ with probability at least 99%.*

We remark that the fact that our algorithm only requires an $\alpha$-truncated sample oracle is what enables us to achieve a polynomial number of oracle calls to the membership oracle to $S$ and significantly improve over the prior works Daskalakis et al. (2018); Lee et al. (2023) (that required super-polynomial oracle calls and samples). For instances where truncated sampling can be done more efficiently than rejection sample we even get logarithmic dependence on the mass of the survival set.

Observe that the algorithm of Theorem 1.5 returns a hypothesis $p_\theta$ whose corresponding truncated density $p_\theta^S$ is close in KL divergence. Typically one asks for the learned density $p_\theta$ to be close to the target non-truncated distribution $p_\theta^*$, that is to extrapolate and match the density beyond the region of observation given by the survival set $S$. We phrased our result in terms of KL divergence of the truncated distributions to keep the required assumptions on the exponential families minimal. In particular, by making the assumptions that render the parameter $\theta^*$ identifiable (as in the prior works) one can readily translate the guarantee of our result to parameter recovery or (learning in KL or total variation) with respect to the target non-truncated density. We next present some new results for well-studied special cases of Definition 1.1 that we obtain as corollaries of Theorem 1.5.

We first present our result for learning truncated Gaussian distributions. A truncated Gaussian density is defined as $\mathcal{N}(x; \mu, \Sigma, S) \propto \mathbb{1}_S(x)\mathcal{N}(x; \mu, \Sigma)$ where $\mu \in \mathbb{R}^d$, is its mean and $\Sigma \in \mathbb{R}^{d \times d}$ is its covariance.

**Corollary 1.6** (Gaussians). *Let $q = \mathcal{N}(\mu^*, \Sigma^*, S)$ be a truncated normal density. There exists an algorithm that draws $M = \frac{d^2}{\epsilon^2}\text{poly}(\log(1/\alpha))$ samples from $q$, makes $M \cdot \text{poly}(1/\alpha)$ oracle calls to the set indicator $\mathbb{1}_S(\cdot)$, runs in $\text{poly}(M, d, 1/\alpha)$ and learns parameters $\hat{\mu}, \hat{\Sigma}$ such that $d_{\text{TV}}(\mathcal{N}(\mu, \Sigma), \mathcal{N}(\hat{\mu}, \hat{\Sigma})) \leq \epsilon$.*

Our next corollary is on the generalization of log-concave exponential families that satify Definition 1.1 and furthermore have log-concave density and polynomial sufficient statistic (see Lee et al. (2023)).

**Corollary 1.7** (Log-Concave Exponential Families). *Fix $\epsilon > 0$. Let $\mathcal{F}$ be a class of log-concave exponential families that satisfy Definition 1.1 and their sufficient statistic $T(x)$ of each $p_\theta \in \mathcal{F}$ is a polynomial of degree $\ell$ that maps $\mathbb{R}^d$ to $\mathbb{R}^k$. Let $q = p_{\theta^*}^S$ be a truncated normal density in $\mathcal{F}$. There exists an algorithm that draws $M = \frac{k}{\epsilon^2}\text{poly}(\log(1/\alpha))$ samples from $q$, makes $M \cdot \text{poly}(1/\alpha)$ oracle calls to the set indicator $\mathbb{1}_S(\cdot)$, runs in $\text{poly}(M, d, 1/\alpha)$, and learns a parameter $\hat{\theta}$ such that $\|\hat{\theta} - \theta^*\|_2 \leq \epsilon$ and $d_{\text{TV}}(p_{\hat{\theta}}, p_{\theta^*}) \leq \epsilon$.*

**Remark:** We refer the reader to Appendix A for a detailed discussion comparing our results in Corollaries 1.6 and 1.7 with the corresponding results in the literature.

## 1.2 OUR TECHNICAL CONTRIBUTION

A common challenge in the line of work on truncated statistics is to ensure that during the optimization phase the parameters of the distribution remain within a region that assigns significantly large mass to the survival set. This is important for many reasons, one being to ensure that gradients remain bounded and another being to ensure that rejection sampling works. As such all prior works restrict the optimization in custom regions of parameters that depend on the class of distributions one is trying to learn. For every different setting, learning Gaussian distributions, product distributions on the hypercube, learning permutations and rankings, and log-concave exponential families, different algorithms have been provided that focus on the specific intricacies of the setting e,g. discrete domain, polynomial features and so on.

Our main technical contribution is offering a unified viewpoint for all those settings as exponential families over an unrestricted domain. Our key technical tool is establishing a very simple region for optimization that ensures the probability of the survival set is bounded and is given directly by the non-truncated loss of the model. More specifically, given an initialization $\theta_0$, we define the following optimization problem that we aim to solve via projected stochastic gradient descent.

$$\min_{\theta \in \Theta} \mathcal{L}_S(\theta) \quad \text{s.t.} \quad \mathcal{L}(\theta) \leq \mathcal{L}(\theta_0) + \log \frac{1}{\alpha}. \tag{3}$$

Given an initial point $\theta^0$ whose loss is sufficiently close to the minimum non-truncated loss $\min_{\theta \in \Theta} \mathcal{L}(\theta)$, any point $\theta$ in the feasible region is guaranteed to assign at least $\Omega(\alpha)$ mass to the survival set.

Beyond our novel understanding of the optimization and sampling landscape, which enables significantly faster convergence, we introduce a much simpler algorithm that makes minimal assumptions about the setting and the distributions involved. Unlike previous works that simultaneously bound sample complexity, query complexity, and parameter closeness, we achieve a tighter analysis by decoupling these terms. Specifically, we demonstrate the following:

- We show that the dependence on samples from the true distribution is limited to the estimation of the mean sufficient statistic, which requires only a few samples. This is possible due to the good concentration properties of the truncated distribution, which behaves as a subexponential distribution.
- We adopt an optimization perspective, focusing on minimizing the given objective as efficiently as possible, rather than directly recovering the underlying parameters. Our guarantees are provided in terms of the closeness between the true truncated density and the learned truncated density.
- Leveraging existing statistical results that relate the closeness of truncated densities to the closeness of parameters and non-truncated densities, we derive parameter recovery algorithms with polynomial dependence on the mass of the survival set.

## 1.3 RELATED WORK

The field of truncated statistics has a long history, which finds its roots in Bernoulli's analysis of smallpox morbidity and mortality data Bernoulli (1760), and the early works of Galton (1897); Pearson (1902a); Pearson & Lee (1908a); Fisher (1931). Thus, we cannot do it justice here and for an overview of the field we refer the reader to Schneider (1986); Cohen (2016); Balakrishnan & Cramer (2014). We also refer the reader to Tobin (1958); Amemiya (1973); Hausman & Wise (1977); Heckman (1979); Maddala (1986); Keane (1993); Hajivassiliou & McFadden (1998), and their references, for further work in statistics and econometrics. There still remain numerous outstanding challenges, targeting density estimation, regression and classification tasks. Many recent works have focused on providing computationally efficient algorithms. There has been a large number of recent works dealing inference with truncated data from a Gaussian distribution Daskalakis et al. (2018); Kontonis et al. (2019), mixtures of Gaussians Nagarajan & Panageas (2019), linear regression Daskalakis et al. (2019); Ilyas et al. (2020); Daskalakis et al. (2020b), sparse Graphical models Bhattacharyya et al. (2021) or Boolean product distributions Fotakis et al. (2020), nonparametric estimation Daskalakis et al. (2021).

The area of robust statistics Huber & Ronchetti (2009) is closely related to our work, as it also addresses the challenge of handling biased datasets with the goal of identifying the underlying data-generating distribution. Recently, there has been substantial theoretical progress in developing computationally efficient methods for robustly estimating high-dimensional distributions in the presence of arbitrary corruptions affecting a small $\varepsilon$ fraction of the samples Diakonikolas et al. (2016); Charikar et al. (2017); Lai et al. (2016); Diakonikolas et al. (2017); Klivans et al. (2018); Diakonikolas et al. (2019). Our work contributes to this broader field of statistical learning, specifically focusing on cases where certain impediments or biases are present.

One of the most notable works in this area is Lee et al. (2023), which introduced efficient algorithms for learning from truncated exponential families. In the following subsection, we provide a detailed comparison between their results and our improved approach.

### 1.3.1 COMPARISON TO LEE ET AL. (2023)

The work of Lee et al. (2023) presents a framework for learning from truncated samples. We outline the key improvements of our method below:

**Oracle Efficiency:** A significant part of our improvement arises from requiring only $\alpha$-truncated sample oracle access (Definition 3.1) throughout our algorithm, whereas Lee et al. (2023) requires access to an $a^{-\log(1/a)}$-truncated sample oracle. Furthermore, any additional improvement on our result is unattainable, as all parameters near $\theta^*$ necessarily require access to an $\alpha$-truncated sample oracle.

**Parameter Estimation:** In Lee et al. (2023), convergence requires approximately $a^{-\log(1/a)}$ iterations, whereas our approach achieves convergence in polynomial time relative to $1/a$. Both Lee et al. (2023) and our work, as shown in Corollary 1.7, have iteration counts that depend on poly(1/mass). However, our advantage stems from being able to identify regions of higher mass more effectively.

**Mass and Sample Complexity:** The guarantees provided in Lee et al. (2023) concern areas of $a^{-\log(1/a)}$ mass, which can lead to inefficiencies. Our approach, by contrast, offers sharper covariance estimates for the truncated distribution, leading to initialization points that are closer to the true truncated parameter $\theta_S^* = \mathbf{E}_{x \sim p_{\theta^*}^S}(T(x))$. While Lee et al. (2023)'s analysis requires remaining within a ball of radius $\log(1/a)$ around the true parameter $\theta^*$, effective learning necessitates both proximity to $\theta^*$ within this radius and maintaining a low negative log-likelihood (NLL) $\mathcal{L}(\theta)$. Had we used the covariance estimates from Lee et al. (2023), our sample complexity would be poly$(1/a)$, but by employing our improved estimates, we achieve poly$(\log(1/a))$, matching the sample complexity seen in Lee et al. (2023).

Initialization and Convergence: While both methods use similar initialization points, our analysis demonstrates faster convergence. Specifically, we show that the initialization point in our method yields a lower starting value for $\mathcal{L}(\theta)$, ensuring the algorithm focuses on high-mass regions more effectively. In contrast, Lee et al. (2023) does not capitalize on this aspect as robustly.

**Summary of Differences:**

- **Sharper Covariance Estimates**: Our approach guarantees that initialization starts within high-mass areas by employing refined covariance estimates.
- **Higher Mass Guarantees**: We emphasize maintaining low NLL values, incrementally broadening the search radius with controlled increases in NLL, which is critical for efficient convergence. This perspective is not as effectively addressed in Lee et al. (2023).

## 2 PRELIMINARIES

A distribution belongs to the **exponential family** if its density is expressed as

$$p_\theta(x) = h(x) \exp(\theta^\top T(x) - A(\theta)).$$

For a given set $S$, we denote by $p_\theta(S)$ the probability that the measure $p_\theta$ assigns to $S$. The function $h(x)$, often referred to as the weight function, is also known by other terms in the literature, such as the carrier measure.

The parameter space of this family is defined as $\Theta = \{\theta \in \mathbb{R}^k : A(\theta) < \infty\}$, which is an open set. After potentially reparameterizing, it is assumed that $\theta$ and $T(x)$ are linearly independent. This form of representation is known as the **minimal representation**. Given a minimal representation, the function $A(\theta)$ is convex. The following properties of an exponential family are standard:

1. The gradient of $A(\theta)$, $\nabla A(\theta)$, is the expected value of the sufficient statistic under the distribution: $\nabla A(\theta) = \mathbf{E}_{x \sim p_\theta}[T(x)]$.
2. The Hessian of $A(\theta)$, $\nabla^2 A(\theta)$, is the covariance of the sufficient statistic under the distribution: $\nabla^2 A(\theta) = \mathbf{Cov}_{p_\theta}[T(x)]$.

We include the definition of a univariate sub-exponential random variable, which can be used to define a corresponding class for the multivariate case by taking the supremum over the unit sphere. There are many equivalent definitions to choose from, but the most convenient for us is by using the moment generating function (MGF). Namely,

**Definition 2.1** (Moment Generating Function). *The moment generating function (MGF) of a distribution $D$, denoted by $M_D(t)$, is defined as*

$$M_D(t) = \mathop{\mathbf{E}}_{x \sim D}[e^{tx}],$$

*provided this expectation exists. The function $M_D(t)$ is defined for all values of $t$ in some interval containing $t = 0$.*

We provide a definition of the sub-exponential distribution, which we will rely on throughout. This definition is similar to the one in Vershynin (2010), with a minor adjustment to account for potential restrictions that arise when scaling moves us outside the natural parameter space $\Theta$ of our model. Note that, aside from some scaling, the definition used here aligns with that in Lee et al. (2023).

**Definition 2.2** (Sub-exponential distribution). *Let $x$ be a univariate random variable with zero mean. The random variable $x$ is said to belong to the class $SE(K_1^2, \beta)$ if the following condition holds: $\mathbf{E}[\exp(\lambda x)] \leq \exp(K_1^2 \lambda^2)$, for all $\lambda$ such that $|\lambda| \leq \frac{1}{\beta}$.*

The following proposition provides an equivalent definition for the sub-exponential class, adapted from Vershynin (2010) with slight modifications to suit our needs. However, since the proof remains unchanged, we omit it.

**Proposition 2.3** (Sub-exponential distribution). *Let $x$ be a univariate random variable with zero mean. Fix parameters $K_i > 0$ for $i = 1, 2$, and $\beta > 0$, such that $x \in SE(K_1^2, \beta)$. Then, the following two properties are equivalent. Additionally, the quantities $\max(K_1, \beta)$ and $K_2$, which appear in the properties, differ by a universal constant.*

*(a) The moment generating function (MGF) of $x$ satisfies*

$$\mathbf{E}[\exp(\lambda x)] \leq \exp(K_1^2 \lambda^2)$$

*for all $\lambda$ such that $|\lambda| \leq \frac{1}{\beta}$.*

*(b) The moments of $x$ satisfy*

$$(\mathbf{E}[|x|^p])^{1/p} \leq K_2 p$$

*for all $p \geq 1$.*

This class of functions is typically associated with a concentration inequality. The one we use, which can also be found in Vershynin (2010), is as follows:

**Fact 2.4** (Bernstein's Inequality). *Let $x_1, \ldots, x_N$ be independent, identically distributed, mean-zero, sub-exponential random variables belonging in the $SE(K_1^2, \beta)$. Then, for every $t \geq 0$, we have*

$$\mathbf{Pr}\left(\left|\frac{1}{N}\sum_{i=1}^N x_i\right| \geq t\right) \leq 2\exp\left(-cN\min\left(\frac{t^2}{\max^2(K_1, \beta)}, \frac{t}{\max(K_1, \beta)}\right)\right),$$

*where $c > 0$ is an absolute constant.*

## 2.1 TRUNCATED STATISTICS

We begin by defining the notion of a **truncated** distribution. Namely, let $p$ be a probability density function on $\mathbb{R}^d$. For a given set $S \subseteq \mathbb{R}^d$, let $p_S$ be the conditional distribution of $x \sim p$ given that $x \in S$, i.e. $p_\theta^S(x) = \frac{p(x) \cdot \mathbb{1}_S(x)}{p_\theta(S)}$. Note that the relative density is $\frac{p_S(x)}{p(x)} = \frac{\mathbb{1}_S(x)}{p(S)}$.

In the context of truncated statistics, we engage with an oracle, denoted by $M_S(x) = \mathbb{1}_S(x)$, which represents the indicator function of the set $S$. This setup is prevalent in scenarios where direct access to the set $S$ is unavailable, yet a mechanism exists to verify the membership of elements within $S$. This method allows for the indirect exploration and analysis of the set through conditional access provided by the oracle.

**Non-truncated and Truncated NLL** Given some exponential family $p_\theta$ parameterized by $\theta \in \mathbb{R}^k$, the negative log-likelihood (NLL) over a population distribution $q$ is defined as follows:

$$\mathcal{L}(\theta) = - \mathop{\mathbf{E}}_{x \sim q} [\log p_\theta(x)] . \qquad (4)$$

In what follows we will refer to $\mathcal{L}$ as "non-truncated". Moreover, for a truncated exponential family $p_\theta^S$, such that the support of $q$ is contained in $S$, we define the following "truncated" version of the NLL objective as:

$$\mathcal{L}_S(\theta) = - \mathop{\mathbf{E}}_{\mathbf{x} \sim q} [\log p_\theta^S(x)] = \mathcal{L}(\theta) + \log p_\theta(S) . \qquad (5)$$

The gradient and Hessian of $\mathcal{L}_S(\cdot)$ are given below.

$$\nabla_\theta \mathcal{L}_S(\theta) = \mathbf{E}_{x \sim p_\theta^S} [T(x)] - \mathbf{E}_{x \sim q} [T(x)] , \quad \nabla_\theta^2 \mathcal{L}_S(\theta) = \mathbf{Cov}_{x \sim p_\theta^S} [T(x)] .$$

## 3 LEARNING TRUNCATED DENSITIES

In this section we prove our main result, namely an efficient algorithm for learning truncated exponential families.

**Definition 3.1** (Sample Oracles). *Given as input an exponential family $p_\theta \in \mathcal{F}$, the (non-truncated) sample oracle returns a sample $x \sim p_\theta$.*

*Moreover, given as input an exponential family $p_\theta \in \mathcal{F}$, the $\beta$-truncated sample oracle over a survival set $S$ returns a sample $\mathbf{x} \sim p_\theta^S$, provided that $p_\theta(S) \geq \beta$. In case this constraint does not hold the behavior of the oracle is undefined.*

**Remark 3.2** (Implementing a $\beta$-Truncated Sample Oracle). *Given a membership oracle to the set $S$ and a sampler for the non-truncated exponential family $p_\theta$, one can directly implement a $\beta$-truncated sample oracle by rejecting samples until one falls in $S$.*

This oracle enables us to reduce the inference problem to an optimization that use the gradients as rewards.

**Theorem 3.3** (Learning Truncated Exponential Families). *Fix $\epsilon > 0$. Let $\mathcal{F}$ be an exponential family with sufficient statistic $T(\cdot)$ such that for any $p_\theta \in \mathcal{F}$, $\lambda I \preceq \mathbf{Cov}_{x \sim p_\theta}[T(x)] \preceq LI$. Assume sample access to an unknown distribution $p_{\theta^*} \in \mathcal{F}$ truncated with a survival set $S$ of mass $\alpha > 0$, and an $\Omega(\alpha^2)$-truncated sample oracle $\mathcal{Q}$ over $S$. There exists an algorithm that uses $\widetilde{O}\left(\frac{k}{\epsilon^2} \log\left(\frac{1}{\alpha}\right)\right)$ samples from $p_{\theta^*}^S$, makes $\widetilde{O}\left(\frac{k}{\epsilon^2} \log\left(\frac{1}{\alpha}\right)^2\right)$ calls to the oracle $\mathcal{Q}$, runs in time $\mathrm{poly}(k, 1/\epsilon, \log(\frac{1}{\alpha}))$ and computes an estimate $\hat{\theta}$, such that $\mathrm{KL}(p_{\theta^*}^S \| p_{\hat{\theta}}^S) \leq \epsilon$ with probability at least 99%.*

**Roadmap to the Proof:** To prove Theorem 3.3, we proceed as follows:

1. **Setup and Definitions:** Our learning algorithm outputs $\hat{\theta}$ by performing PSGD (Theorem 3.12) starting from an initial point $\theta_0$ over a domain $D$. The key properties are that $\theta^* \in D$ and, for all $\theta \in D$, the bound $\mathbf{Pr}_{x \sim p_\theta}[x \in S] \geq c\alpha^{-2}$ holds.
2. **Main Argument:** The parameter $\theta_0$ serves as a good approximation of $\mathbf{E}_{x \sim p_{\theta^*}^S}[T(x)]$. This approximation is achieved through a collection of smoothness results, specifically Lemma 3.5, Lemma 3.11, and Lemma 3.7. This good initialization is crucial for defining $D = \{\theta : \mathcal{L}(\theta) - \mathcal{L}(\theta_0) \leq \log(1/\alpha)\}$ with the necessary properties. As shown in Corollary 3.9, initialization from a point with sufficiently low loss $\mathcal{L}(\theta_0) \leq \mathcal{L}(\theta^0) + \epsilon$ allows the application of Corollary 3.10, where $\theta^0$ is defined in Corollary 3.10.
3. **PSGD Specifications:** We apply Theorem 3.12 with $f = \mathcal{L}_S$. The smoothness parameter $L_S$ is $L(1 + \log^2(1/\alpha))$ over the domain $D$, owing to the smoothness of the truncation (Lemma 3.5) and the sufficient mass within $D$. Similarly, as shown in Lemma 3.13, the gradient variance of $\mathcal{L}_S$ is of order $kL \log^6(1/\alpha)$ on $D$.

Using the same techniques as in Theorem 3.3, we can demonstrate that an $\Omega(\alpha)$-truncated sample oracle $\mathcal{Q}$ over $S$ is feasible. Note that since the parameter $\theta^*$ assigns mass $\alpha$ to $S$, we cannot hope to achieve something more efficient than an $\Omega(\alpha)$-truncated sample oracle. The main idea for this improvement in Theorem 3.3 is to incrementally expand the exploration domain to avoid overshooting and inadvertently ending up in a low-mass region. We include the theorem statement below; for a detailed proof, along with the necessary apparatus, look at the appendix Appendix C.

**Theorem 3.4** (Learning Truncated Exponential Families ). *Fix $\epsilon > 0$. Let $\mathcal{F}$ be an exponential family with sufficient statistic $T(\cdot)$ such that for any $p_\theta \in \mathcal{F}$, $\lambda I \preceq \mathbf{Cov}_{x \sim p_\theta}[T(x)] \preceq LI$. Assume sample access to an unknown distribution $p_{\theta^*} \in \mathcal{F}$ truncated with a survival set $S$ of mass $\alpha > 0$, and an $\Omega(\alpha)$-truncated sample oracle $\mathcal{Q}$ over $S$. There exists an algorithm that uses $\widetilde{O}\left(\frac{k}{\epsilon^2}\log\left(\frac{1}{\alpha}\right)\right)$ samples from $p_{\theta^*}^S$, makes $\widetilde{O}\left(\frac{k}{\epsilon^2}\log\left(\frac{1}{\alpha}\right)^2\right)$ calls to the oracle $\mathcal{Q}$, runs in time $\mathrm{poly}(k, 1/\epsilon, \log(\frac{1}{\alpha}))$ and computes an estimate $\hat{\theta}$, such that $\mathrm{KL}(p_{\theta^*}^S \| p_{\hat{\theta}}^S) \leq \epsilon$ with probability at least 99%.*

### 3.1 FINDING A GOOD INITIALIZATION

To demonstrate that a good initialization can be found with $\widetilde{O}\left(\frac{k}{\epsilon^2}\log\left(\frac{1}{\alpha}\right)\right)$ samples from $p_{\theta^*}^S$, we leverage the concentration properties of the distribution $p_{\theta^*}^S$. This is a threefold process, and our end goal is to find in which sub-exponential class $x \sim p_{\theta^*}^S$ belongs. In order to accomplish that, we have to bound $\mathbf{Cov}_{x \sim p_{\theta^*}^S}(T(X))$, and in order to do that we need $\|\mathbf{E}_{p_\theta^S}[T(x)] - \mu\|_2 \lesssim \log\left(\frac{1}{p_\theta(S)}\right)$. The next lemma will deal with those two latter stepping stones.

**Lemma 3.5** (Moment preservation after truncation). *Assume that $\mathbf{E}_{x \sim p_\theta}[T(x)] = \mu$, $\mathbf{Cov}_{x \sim p_\theta}(T(x)) \preceq LI$, and $p_\theta(S) > 0$. Then, $\|\mathbf{E}_{x \sim p_\theta^S}[T(x)] - \mu\|_2 \lesssim \log\left(\frac{1}{p_\theta(S)}\right)$. Similarly, we have the following covariance estimate: $\mathbf{Cov}_{x \sim p_\theta^S}(T(x)) \preceq \left(O\left(\log^2\left(\frac{1}{p_\theta(S)}\right)\right) + L\right)I$.*

The previous lemma gives us a bound on how much the smoothness is affected by the truncation. Hence, tighter bounds guarantee needing fewer samples since the truncated distribution enjoys stronger concentration properties. The analysis is achieved by performing a worst-case analysis.

Next, we see how truncation affects the sub-exponential property that the non-truncated distribution possesses. For the proof we refer to Appendix B.

**Lemma 3.6** (Truncated density is sub-exponential). *Let $\mathcal{F}$ be an exponential family with sufficient statistic $T(\cdot)$ such that for any $p_\theta \in \mathcal{F}$, we have $\lambda I \preceq \mathbf{Cov}_{x \sim p_\theta}[T(x)] \preceq LI$. Let $x$ follow the truncated distribution: $p_\theta^S(x) = \frac{p_\theta(x)\mathbb{1}_S(x)}{p_\theta(S)}$, where $p_\theta(x) = h_S(x)\exp(\theta^\top T(x) - A_S(\theta))$ and $p_\theta(S)$ is the normalization constant. Then, the random variable $T(x)$, under this truncated distribution, is sub-exponential, denoted $SE(K^2, \beta)$, with parameters $K^2 = (1 + \log^2(1/\alpha))L$ as $K$ appears in Definition 2.2, and $\beta$ is the reciprocal of the largest radius $r$ of the ball $B(\theta, r)$, centered at $\theta$, that is contained in $B(\theta, r) \subset \Theta$.*

Now, we have developed all the tools necessary to find a sharp estimate of the samples needed for the empirical truncated distribution to approximate its mean.

**Lemma 3.7** (Empirical estimation of the truncated sufficient statistic). *Fix $\epsilon > 0$. Let $p_{\theta^*}$ be some exponential family such that $p_{\theta^*}(S) = \alpha$. Moreover, let $\theta_S^* = \mathbf{E}_{x \sim p_{\theta^*}^S}[T(x)]$ be the expected sufficient statistic of the truncated exponential family $p_{\theta^*}^S$. Moreover, let $\hat{\theta} = \frac{1}{N}\sum_{i=1}^N T(x^{(i)})$ be the corresponding empirical statistic for $x^{(1)}, \ldots, x^{(N)}$ i.i.d. samples from $p_{\theta^*}^S$. Using $N = O\left(\frac{kL}{\epsilon^2}\log\left(\frac{1}{\delta}\right)\log^2\left(\frac{1}{\alpha}\right)\right)$ samples, with probability at least $1 - \delta$, it holds $\|\hat{\theta} - \theta_S^*\|_2 < \epsilon$.*

*Proof.* Our estimator is $\hat{\theta} = \frac{1}{N}\sum_{i=1}^N T(x_i)$, where $x_i \sim p_{\theta^*}^S$. We have shown that, for any unit vector $u$, $Y_i = u^T T(x_i) - u^T \mathbf{E}_{x_i \sim p_{\theta^*}^S}[T(x_i)]$ belongs in the sub-exponential class with $K^2 = O\left(L + \log^2\left(\frac{1}{\alpha}\right)\right)$ We fix $t = \epsilon > 0$, now by using Bernstein's inequality, as seen in Fact 2.4, we obtain

$$p_{\theta^*}^S\left(\left|\frac{1}{N}\sum_{i=1}^N Y_i\right| \geq t\right) \leq 2\exp\left(-cN\min\left(\frac{t^2}{\max^2(K, \beta)}, \frac{t}{\max(K, \beta)|}\right)\right) = 2\exp\left(-c\frac{N\epsilon^2}{\max^2(K, \beta)}\right)$$

Since the previous inequality holds for all $u$, and by the superposition property of sub-exponential random variables, we obtain the previously mentioned bound for the norm, namely:

$$p_{\theta^*}^S \left( \left\| \hat{\theta} - \mathbf{E}_{x_i \sim p_{\theta^*}^S} \left( T(x_i) \right) \right\| \geq t \right) \leq 2 \exp \left( -c \frac{N \epsilon^2}{\max^2(K, \beta)} \right)$$

Since $b, L$ is a constant independent of $\alpha$, we have the following bound: $\max^2(K, \beta) = O\left( \log^2 \left( \frac{1}{\alpha} \right) \right)$. Therefore, for $N \geq O\left( \frac{k}{\epsilon^2} \log^2 \left( \frac{1}{\alpha} \right) \log \left( \frac{1}{\delta} \right) \right)$, we get, $p_{\theta^*}^S \left( \left\| \hat{\theta} - \mathbf{E}_{x_i \sim p_{\theta^*}^S} \left( T(x_i) \right) \right\| \geq t \right) \leq \delta$. $\qquad \square$

## 3.2 THE FEASIBLE REGION AROUND INITIALIZATION

To ensure feasiblity we will use the following simple observation.

**Observation 3.8** (Mass along the sub-level sets)**.** *Let $p_\theta$ be some exponential family and define $\theta'$ to be a minimizer of the constrained objective $\{\min_\theta \mathcal{L}_S(\theta) : s.t. \mathcal{L}(\theta) \leq \bar{L}\}$ for some $\bar{L}$. For any $\theta$ such that $\mathcal{L}(\theta) \leq \bar{L}$, it holds $\mathbf{Pr}_\theta[S] \geq \mathbf{Pr}_{\theta'}[S] \exp(\mathcal{L}(\theta') - \mathcal{L}(\theta))$.*

*Proof.* This holds as $\mathcal{L}_S(\theta') \leq \mathcal{L}_S(\theta)$ which directly gives the required bound. $\qquad \square$

Using Observation 3.8 we readily obtain the following corollary.

**Corollary 3.9** (Mass monotonicity along sub-level thresholds)**.** *Let $\theta^*$ be the minimizer of the truncated NLL objective $\mathcal{L}_S$ for some exponential family $p_\theta$. Denote by $\alpha = \mathbf{Pr}_{\theta^*}[S]$ the mass assigned to the survival set $S$. Define the following constrained optimization problem: $\{\min_\theta \mathcal{L}_S(\theta) : s.t. \mathcal{L}(\theta) \leq \bar{L}\}$ and denote by $\theta_1$ its solution with $\bar{L} = L_1$ and by $\theta_2$ its solution of the same optimization problem with $\bar{L} = L_2$, where $L_2 > L_1$.*

1. *(Monotonicity) The mass assigned to $S$ is decreasing as a function of $\bar{L}$: $\mathbf{Pr}_{\theta_1}[S] \geq \mathbf{Pr}_{\theta_2}[S] \geq \alpha$.*
2. *(Exponential decrease) The mass assigned to $S$ by $\theta_2$ drops exponentially fast, i.e., it holds that $\mathbf{Pr}_{\theta_2}[S] \leq \mathbf{Pr}_{\theta_1}[S] \, e^{-(L_2 - L_1)}$.*

*Proof.* Notice that by definition of $\theta_1$, $\theta_1 \in \{\min_\theta \mathcal{L}_S(\theta) : s.t. \mathcal{L}(\theta) \leq L_2\}$. Hence, $\mathbf{Pr}_{\theta_1}[S] \geq \mathbf{Pr}_{\theta_2}[S] \exp(L_2 - L_1) \geq \mathbf{Pr}_{\theta_2}[S]$. It only remains to show that $\mathbf{Pr}_{\theta_2}[S] \geq \alpha$. Since, the $\{\min_\theta \mathcal{L}_S(\theta) : s.t. \mathcal{L}(\theta) \leq L_2\}$ decreases as $L_2$ increases, $\mathcal{L}(\theta^*) = \{\min_\theta \mathcal{L}_S(\theta) : s.t. \mathcal{L}(\theta) < \infty\} = \min\{\mathcal{L}_S(\theta) : s.t. \mathcal{L}(\theta) \leq \mathcal{L}(\theta^*)\}$. Which means for $\theta'$ that is the solution to a constrained optimization problem with lower $\bar{L} = L_2$, we have $\theta' \in \{\mathcal{L}_S(\theta) : s.t. \mathcal{L}(\theta) \leq \mathcal{L}(\theta^*)\}$. Hence, $\bar{L} \leq \mathcal{L}(\theta^*)$ and from the previous part this implies $\alpha = \mathbf{Pr}_{\theta^*}[S] \leq \mathbf{Pr}_{\theta'}[S]$. Hence, we conclude. $\qquad \square$

Given the exponential decrease, i.e. Corollary 3.9 we obtain the following corollary. Which is ensuring that our initialization of PSGD is close in LLN distance, and that all our parameters inside our projection domain give $\mathrm{poly}(1/\alpha)$ to the survival set $S$.

**Corollary 3.10** (Feasibility of $\theta^0$)**.** *Suppose that $\theta^0$ is $\theta^0 = \mathbf{E}_{x \sim p_{\theta^*}^S}[T(x)]$. Then, $\mathcal{L}(\theta^*) \leq \mathcal{L}(\theta^0) + \log\left( \frac{1}{\alpha} \right)$. Furthermore, if for any $\theta$ such that $\mathcal{L}(\theta) \leq \mathcal{L}(\theta^0) + \log\left( \frac{1}{\alpha} \right) + \epsilon$, then $p_\theta(S) \geq p_{\theta^0}(S) \alpha e^{-\epsilon} \geq \alpha^2 e^{-\epsilon}$.*

*Proof.* From Corollary 3.9 it holds that $p_{\theta^0}(S) > \alpha$. Hence, by the exponential decrease, provided by the same corollary, we find that $\mathcal{L}(\theta^*) \leq \mathcal{L}(\theta^0) + \log\left( \frac{1}{\alpha} \right)$. The second part is an immediate consequence of the Observation 3.8. $\qquad \square$

## 3.3 PSGD CONVERGENCE

We prove the following lemma show that the truncated densities are sub-exponential, see Appendix B.

**Lemma 3.11** (Sub-exponential Property for the Truncated Distribution)**.** *Let $x$ follow the truncated distribution: $p_\theta^S(x) = \frac{p_\theta(x) \mathbb{1}_S(x)}{p_\theta(S)}$, where $p_\theta(x) = h(x) \exp(\theta^\top T(x) - A(\theta))$ and $\mathbf{Pr}_{x \sim p_\theta}[x \in S]$*

*is the normalization constant. Then the random variable $T(x)$ under this truncated distribution is sub-exponential. More specifically, it belongs to the family $SE(K^2, \beta)$ where $K^2 = L + \log^2\left(\frac{1}{\alpha}\right)$, $\beta^{-1} = \inf_{\|u\|=1} \sup\{\gamma : \gamma u + \theta \in \Theta\}$.*

The result for PSGD we will use is the following. Its proof is standard, see, .e.g, Shalev-Shwartz & Ben-David (2014) or Appendix B for a proof.

**Theorem 3.12** (Convergence of Projected Stochastic Gradient Descent). *Consider the Projected Stochastic Gradient Descent (PSGD) algorithm for minimizing a convex function $f$ over a convex set $\mathcal{K}$. Assume the following:*

1. ***Lipschitz Continuous Gradients***: *There exists a constant $M > 0$ such that for all $x$ and $y$, $\|\nabla f(x) - \nabla f(y)\| \le M\|x - y\|$.*
2. ***Bounded Variance***: *The variance of the stochastic gradients is bounded, i.e., there exists a constant $b^2$ such that $\mathbf{E}[\|g^{(t)}\|^2] \le b^2$, where $g^{(t)} = \nabla f(x_t)$.*

*Then, for $\tilde{w} = \frac{1}{T}\sum_{i=1}^{T} w^{(i)}$ where the $\{w^{(i)}\}$ are generated by the PSGD algorithm with step size $h_t = \frac{1}{Mt}$, it holds $\mathbf{E}\left[f(\tilde{w})\right] - f(w^*) \le \frac{\|w^{(0)} - w^*\|^2}{2MT} + \frac{b^2\pi^2}{12M^2T}$.*

To apply Theorem 3.12 we first show that our unbiased gradients have bounded second moment. The proof can be found in Appendix B.

**Lemma 3.13** (Bounded variances of stochastic gradients ). *Let $T(z) : \mathbb{R}^d \mapsto \mathbb{R}^k$ be the sufficient statistic of a truncated exponential family $p_{\theta*}^S$. We have that $v = T(z) - T(x)$, where $z \sim p_\theta^S$ and $x \sim p_{\theta*}^S$ is an unbiased estimator of $\nabla\mathcal{L}_S(\theta)$. Moreover, $v$ has bounded second moment, namely,*
$$\mathbf{E}[\|v\|^2] = \mathbf{E}_{z \sim p_\theta^S} \mathbf{E}_{x \sim p_{\theta*}^S}\left[\|T(z) - T(x)\|^2\right] \lesssim kL\log^6(1/\alpha).$$

### 3.4 The Proof of Theorem 3.3

We are now ready to prove our main result. Apply Theorem 3.12 (PSGD) to the optimization problem Equation (3) for $f = \mathcal{L}_S$ and $w^{(0)} = \theta_0$, with projecting region $D = \{\theta : \mathcal{L}(\theta) \le \mathcal{L}(\theta_0) + \log\frac{1}{\alpha}\}$. Here, $\theta_0 = \frac{1}{N}\sum_{i=1}^{N} T(x_i)$ for $N \ge O\left(\frac{kL^3}{\epsilon^2}\log\left(\frac{1}{\delta}\right)\log^2\left(\frac{1}{\alpha}\right)\right)$. We also set $\theta^0 = \mathbf{E}_{x \sim p_{\theta*}^S}[T(x)]$. Then, by applying Lemma 3.7 with $\epsilon = \frac{\epsilon}{L}$, we obtain $\|\theta_0 - \theta^0\|_2 \le \frac{\epsilon}{L}$. Observe now, that by the $L$ smoothness of $\mathcal{L}$ and the fact that $\nabla\mathcal{L}(\theta^0) = 0$, we have the bound $\|\nabla\mathcal{L}(\theta)\| \le L \cdot r$ for all $\theta \in \{\theta : \|\theta - \theta^0\| \le r\}$, hence $|\mathcal{L}(\theta_0) - \mathcal{L}(\theta^0)| \le \epsilon$.

Furthermore, by Corollary 3.10 we get $\theta^* \in D \subset \{\theta : \mathcal{L}(\theta) - \mathcal{L}(\theta^0) \le \log(\frac{1}{\alpha}) + \epsilon\}$, and $p_\theta(S) \ge \alpha^2 e^{-\epsilon}$ for all $\theta \in D$. By strong convexity of $\mathcal{L}(\theta)$, we have that $\frac{\lambda}{2}\|\theta^0 - \theta^*\|_2^2 \le O\left(\log\left(\frac{1}{\alpha}\right)\right)$. Hence, Theorem 3.12 in our setup gives an upper bound $\frac{C\log(1/\alpha)}{2(L+\log^2(1/\alpha))T} + \frac{b\pi^2}{12(L+\log^2(1/\alpha))^2T}$. So, for $T \ge \frac{1}{\epsilon}\left(\frac{6C\log^2(1/\alpha)(L+\log^2(1/\alpha))^2 + \pi^2}{12(L+\log^2(1/\alpha))^2}\right)$, Theorem 3.12 gives us an $\overline{\theta}$ such that $\mathbf{E}\left[f(\overline{\theta})\right] - f(\theta^*) \le \epsilon$. From Markov's inequality, we get $\mathbf{Pr}\left[f(\overline{\theta}) - f(\theta^*) \ge 3\epsilon\right] \le \frac{1}{3}$

We can easily amplify this probability by repeating this process independently and hence obtaining a sequence of $\overline{\theta}_1, \overline{\theta}_2, \ldots, \overline{\theta}_m$ and then choosing $\overline{\theta} = \arg\min_{\overline{\theta}_i} f(\overline{\theta}_i)$. Set $\hat{\theta} := \overline{\theta}$ and observe that $\mathbf{Pr}\left[f(\hat{\theta}) - f(\theta^*) \ge 3\epsilon\right] \le \left(\frac{1}{3}\right)^m$. Hence by choosing $m \ge \log(\delta)/\log(1/3)$, we obtain a $\hat{\theta}$ that satisfies $\mathbf{Pr}\left[f(\hat{\theta}) - f(\theta^*) \ge 3\epsilon\right] \le \delta$. To accomplish this, however, we need access to the value of $\mathcal{L}_S(\overline{s}_i) = \mathcal{L}(\overline{s}_i) + \log(\mathbf{Pr}_{\overline{s}_i}[S])$. Since we have access to $S$ only through its oracle, in order to calculate $\mathbf{Pr}_{\overline{w}_i}[S]$, we use concentration for a Bernoulli random variable. More specifically, by Hoeffding's inequality, we need $O\left(\frac{1}{\epsilon}\log\left(\frac{1}{\alpha}\right)\right)$ samples to estimate $\log\left(\mathbf{Pr}_{\overline{w}_i}[S]\right)$, $\epsilon$-close with probability at least $1 - \delta$.

Therefore, we obtain a $\hat{\theta}$ with high probability that achieves high precision, namely $\mathcal{L}_S(\hat{\theta}) - \mathcal{L}_S(\theta^*) < \epsilon$ or equivalently $\mathrm{KL}(p_{\theta*}^S \| p_{\hat{\theta}}^S) < \epsilon$.

## ACKNOWLEDGMENTS

This work has been partially supported by project MIS 5154714 of the National Recovery and Resilience Plan Greece 2.0, funded by the European Union under the NextGenerationEU Program.

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

## A  COMPARISON WITH PREVIOUS WORK

In this appendix, we present the main results from prior research on learning truncated distributions, specifically those Daskalakis et al. (2018) on truncated Gaussian distributions and Lee et al. (2023) on truncated exponential families. We then compare these results with ours to highlight improvements in oracle complexity and dependence on the truncation parameter $\alpha$.

### A.1  RESULTS FROM DASKALAKIS ET AL. (2018)

Daskalakis et al. (2018) investigated the problem of learning a Gaussian distribution truncated to a measurable set. Their main result can be summarized as follows:

**Theorem A.1** (Learning Truncated Gaussians Daskalakis et al. (2018)). *Let $q = \mathcal{N}(\mu^*, \Sigma^*, S)$ be a Gaussian distribution truncated to a measurable set $S \subseteq \mathbb{R}^d$, with mean $\mu^* \in \mathbb{R}^d$ and covariance $\Sigma^* \in \mathbb{R}^{d \times d}$. Set $\alpha = \mathbb{P}_{x \sim \mathcal{N}(\mu^*, \Sigma^*)}[x \in S]$. Then, there exists an algorithm that:*

- *Requires a number of samples $M = \frac{d^2}{\epsilon^2}\text{poly}(\log(1/\alpha))$ and makes $M \cdot \alpha^{-\text{poly}(\log(1/\alpha))}$ oracle calls to the set indicator $\mathbb{1}_S(\cdot)$.*
- *Runs in $\text{poly}(M, d, \alpha^{-\text{poly}(\log(1/\alpha))})$.*

*The algorithm computes estimates $\hat{\mu}$ and $\hat{\Sigma}$ such that, with high probability,*

$$d_{\text{TV}}(\mathcal{N}(\hat{\mu}, \hat{\Sigma}), \mathcal{N}(\mu^*, \Sigma^*)) < \epsilon$$

The super-polynomial dependence on $\frac{1}{\alpha}$ arises because the mass assigned to the truncation set $S$ within the optimization domain used in their analysis becomes super-polynomially small.

### A.2  RESULTS FROM LEE ET AL. (2023)

Lee et al. (2023) extended the analysis to general exponential families truncated to a measurable set $S$. Their main result is as follows:

**Theorem A.2** (Learning Truncated Exponential Families Lee et al. (2023)). *Let $\mathcal{F}$ be a class of log-concave exponential families that satisfy Definition 1.1 and their sufficient statistic $T(x)$ of each $p_\theta \in \mathcal{F}$ is a polynomial of degree $\ell$ that maps $\mathbb{R}^d$ to $\mathbb{R}^k$. Suppose $p_{\theta^*}$ is the target distribution, and $q = p_{\theta^*}^S$ is its truncation to a set $S$. Set $\alpha = \mathbb{P}_{x \sim p_{\theta^*}}[x \in S]$. Then, there exists an algorithm that:*

- *Requires a number of samples $M = \frac{k}{\epsilon^2}\text{poly}(\log(1/\alpha))$ and makes $M \cdot \text{poly}(\alpha^{-\log(1/\alpha)})$ oracle calls to the set indicator $\mathbb{1}_S(\cdot)$.*
- *Runs in $\text{poly}(M, k, \alpha^{-\log(1/\alpha)})$.*

*The algorithm outputs an estimate $\hat{\theta}$ such that, with high probability, $d_{\text{TV}}(p_{\hat{\theta}}, p_{\theta^*}) \leq \epsilon$.*

Again, the super-polynomial dependence on $\frac{1}{\alpha}$ is due to the small probability mass assigned to $S$, which adversely impacts the optimization landscape.

### A.3  COMPARISON AND DISCUSSION

The primary difference between these prior results and our work lies in the dependence on the truncation parameter $\alpha$. In both Daskalakis et al. (2018) and Lee et al. (2023), the algorithms require a number of oracle calls, and computational time that are super-polynomial in $\frac{1}{\alpha}$. This is mainly because the probability mass assigned to the truncation set $S$ as it becomes small it negatively affects the smoothness and strong convexity properties needed for efficient optimization.

In contrast, our algorithms achieve similar estimation guarantees while requiring a number of calls to the indicator of the truncation set $\mathbb{1}_S$ that depends polynomially on $\frac{1}{\alpha}$. Specifically, we ensure that for all parameters $\theta$ in the domain $D$ of our algorithm, the probability mass assigned to $S$ satisfies

$$\mathbb{P}_{x \sim p_\theta}[x \in S] \geq \text{poly}(\alpha).$$

This lower bound prevents the mass from becoming negligibly small as $\alpha$ decreases. Consequently, the strong convexity constant of the truncated negative log-likelihood $\mathcal{L}_S(\theta)$ (Equation (3)) depends polynomially on $\alpha$, enabling efficient optimization.

## A.4 CONCLUSION

By carefully analyzing the distribution of the mass assigned to the truncation set $S$, we have developed algorithms that reduce the dependence on $\alpha$ from super-polynomial to polynomial in $\frac{1}{\alpha}$. This improvement makes our methods more efficient for small values of $\alpha$ by optimizing the computational landscape and reducing the complexity of implementing the required oracle.

## B OMMITED PROOFS

### B.1 THE PROOF OF LEMMA 3.11

The truncated exponential family distribution can be written as:

$$p_\theta^S(x) = h_S(x) \exp(\theta^\top T(x) - A_S(\theta)),$$

where:

- $h_S(x) = h(x)1_S(x)$ is the modified base measure, zero outside the set $S$,
- $A_S(\theta) = A(\theta) + \log p_\theta(S)$ is the modified log-partition function reflecting truncation.

Define the truncated log-partition function:

$$A_S(\theta) = \log \int_S h(x) \exp(\theta^\top T(x)) \, dx.$$

The expected value and covariance under truncation are given by:

$$\mu_S = \nabla A_S(\theta),$$

$$\mathbf{Cov}_{t \sim p_\theta^S}(T(x)) = \nabla^2 A_S(\theta).$$

Consider the moment generating function (MGF): To simplify the calculations deal with the pushforward measure induced by the mapping $x \to T(x)$ and we denote the corresponding density as $p_\theta^S(t) = h_S(t) \exp(\theta t - A_S(\theta))$.

$$E_{t \sim p_\theta^S}[e^{\gamma u^\top (t - \mu_S)}] = e^{-\gamma u^\top \mu_S} \frac{Z_S(\gamma u + \theta)}{Z_S(\theta)},$$

and to establish subexponentiality, we need to establish an inequality of the form:

$$\frac{Z_S(\gamma u + \theta)}{Z_S(\theta)} \cdot e^{-\gamma u^\top \mu_S} \le e^{\gamma^2 K^2}$$

for $\gamma$ such that both $\theta + \gamma u$ and $\theta + \gamma u$ belong in our parameter space. Using the $\left(L + \log^2\left(\frac{1}{\alpha}\right)\right)$-smooth property of $A_S(\theta)$ we find:

$$A_S(\gamma u + \theta) - A_S(\theta) \le \gamma u^\top \mu_S + \left(L + \log^2\left(\frac{1}{\alpha}\right)\right)\gamma^2.$$

Hence,

$$E_{t \sim p_\theta^S}[e^{\gamma u^\top (t - \mu_S)}] = e^{-\gamma u^\top \mu_S} \frac{Z_S(\gamma u + \theta)}{Z_S(\theta)} \le \exp\left(\gamma^2\left(L + \log^2\left(\frac{1}{\alpha}\right)\right)\right)$$

Therefore, $T(x)$ is $SE(K^2, \beta)$ with

$$K^2 = \left(L + \log^2\left(\frac{1}{\alpha}\right)\right), \ \beta^{-1} = \min\left(\sup\{\gamma : \gamma u + \theta \in \Theta\}, \sup\{\gamma : -\gamma u + \theta \in \Theta\}\right).$$

## B.2 THE PROOF OF LEMMA 3.5

First, for convenience, suppose that $p_\theta(S) = \alpha$. To establish the bound, we will show that for any vector $\mathbf{u} \in \mathbb{R}^d$, the following holds:

$$\left| \mathbf{E}_{x \sim p_\theta^S}[u^T T(x)] - \mathbf{E}_{x \sim p_\theta}[u^T T(x)] \right| \leq O\left(\log\left(\frac{1}{\alpha}\right)\right).$$

Indeed, for all vectors $w$ and all norms $\|\cdot\|$, there is a linear functional $f$ such that $f(w) = \|w\|$. Since every linear functional is of the form $w \to u^T \cdot w$, we can be sure that, after choosing the proper $u$, we can get a bound for any norm that may be of use.

Define $C_R = \{x : u^T T(x) > R\}$. We change the measure in the first expectation, so we can compare them more readily, namely,

$$\left| \mathbf{E}_{x \sim p_\theta^S}[u^T T(x)] - \mathbf{E}_{x \sim p_\theta}[u^T T(x)] \right|$$

$$= \left| \mathbf{E}_{x \sim p_\theta}\left[ \frac{p^S}{p} u^T T(x) \right] - \mathbf{E}_{x \sim p_\theta}\left[ u^T T(x) \right] \right|$$

$$= \left| \mathbf{E}_{x \sim p_\theta}\left[ \left(\frac{\mathbf{1}_S - \alpha}{\alpha}\right) u^T T(x) \right] \right|$$

$$= \left| \mathbf{E}_{x \sim p_\theta}\left[ \left(\frac{\mathbf{1}_S - \alpha}{\alpha}\right) u^T T(x) \mathbf{1}_{C_R} \right] + \mathbf{E}_{x \sim p_\theta}\left[ \left(\frac{\mathbf{1}_S - \alpha}{\alpha}\right) u^T T(x) \mathbf{1}_{C_R^c} \right] \right|$$

$$\leq \left| \mathbf{E}_{x \sim p_\theta} \left| \left(\frac{\mathbf{1}_S - \alpha}{\alpha}\right) u^T T(x) \mathbf{1}_{C_R} \right| + \left| \mathbf{E}_{x \sim p_\theta}\left[ \left(\frac{\mathbf{1}_S - \alpha}{\alpha}\right) R \right] \right| \right|$$

$$\leq \left| \mathbf{E}_{x \sim p_\theta}\left[ \left(\frac{\mathbf{1}_S - \alpha}{\alpha}\right) u^T T(x) \mathbf{1}_{C_R} \right] \right| + \left| \mathbf{E}_{x \sim p_\theta}\left[ \left(\frac{\mathbf{1}_S - \alpha}{\alpha}\right) R \right] \right|$$

$$\leq \frac{1}{\alpha} \mathbf{E}_{x \sim p_\theta}\left[ u^T T(x) \mathbf{1}_{C_R} \right] + \mathbf{E}_{x \sim p_\theta}\left[ \left|\frac{\mathbf{1}_S - \alpha}{\alpha}\right| R \right]$$

$$\leq \frac{1}{\alpha} \mathbf{E}_{x \sim p_\theta}\left[ u^T T(x) \mathbf{1}_{C_R} \right] + R$$

It remains to work on the term $\mathbf{E}_{x \sim p_\theta}\left[ u^T T(x) \mathbf{1}_{C_R} \right]$:

$$\mathbf{E}_{x \sim p_\theta}\left[ u^T T(x) \mathbf{1}_{C_R} \right] = \mathbf{E}_{x \sim p_\theta}\left[ u^T T(x) | C_R \right] p_\theta(C_R)$$

$$= \mathbf{E}_{x \sim p_\theta}\left[ \log(\exp u^T T(x) | C_R) \right] p_\theta(C_R)$$

$$= \log\left( \mathbf{E}_{x \sim p_\theta}[\exp u^T T(x) | C_R] \right) p_\theta(C_R)$$

$$\leq \log\left( \mathbf{E}_{x \sim p_\theta}[\exp u^T T(x) \mathbf{1}_{C_R}] \frac{1}{p_\theta(C_R)} \right) p_\theta(C_R)$$

$$\leq \log\left( \mathbf{E}_{x \sim p_\theta}[\exp u^T T(x) \mathbf{1}_{C_R}] \frac{1}{p_\theta(C_R)} \right) p_\theta(C_R)$$

$$\leq \log\left( \mathbf{E}_{x \sim p_\theta}[\exp u^T T(x) \mathbf{1}_{C_R}] \right) p_\theta(C_R) - \log\left( p_\theta(C_R) \right) p_\theta(C_R)$$

We switch our focus to the term $\log\left( \mathbf{E}_{x \sim p_\theta}[\exp u^T T(x) \mathbf{1}_{C_R}] \right)$

$$\log\left( \mathbf{E}_{x \sim p_\theta}[\exp u^T T(x) \mathbf{1}_{C_R}] \right) \leq \frac{1}{2} \log\left( \mathbf{E}_{x \sim p_\theta}\left[ \exp(2u^T T(x)) \right] \mathbf{E}_{x \sim p_\theta}[\mathbf{1}_{C_R}] \right)$$

$$\leq \frac{1}{2} \log\left( \mathbf{E}_{x \sim p_\theta}\left[ \exp(2u^T T(x)) \right] \right) \cdot \log\left( \mathbf{E}_{x \sim p_\theta}[\mathbf{1}_{C_R}] \right)$$

$$\leq \frac{1}{2} \log\left( \mathbf{E}_{x \sim p_\theta}\left[ \exp(2u^T T(x)) \right] \right) \cdot \log\left( p_\theta(C_R) \right)$$

$$\leq \frac{1}{2} \log\left( \exp\left( 4L + 2u^T \mathop{\mathbf{E}}_{x \sim p_\theta}[T(x)] \right) \right) \cdot \log\left( p_\theta(C_R) \right)$$

$$\leq \frac{1}{2} \left( 4L + 2u^T \mathbf{E}_{x \sim p_\theta}[T(x)] \right) \cdot \log\left( p_\theta(C_R) \right)$$

For convenience, we set $p_\theta(C_R) = c(R)$, and by putting everything together, we obtain

$$\left| \mathbf{E}_{x \sim p_\theta^S}[u^T T(x)] - \mathbf{E}_{x \sim p_\theta}[u^T T(x)] \right| \le \frac{1}{2\alpha} \left( 4L + 2u^T \mathbf{E}_{p_\theta}[T(x)] \right) \cdot \log\left( c(R) \right) c(R)$$

$$- \frac{1}{\alpha} \log(c(R)) \cdot c(R) + R$$

Since $u^T T(x)$ is subexponential, we have that $p_\theta(C_R) \le C \exp(-cLR)$. Where the constant $C = O\left( \log \mathbf{E}_{x \sim p_\theta}[p(x)] \right)$, which is paid since $p(x)$ is not centered. Substituting back into the original inequality, we get a bound of the form

$$O\left( \frac{1}{\alpha} \exp\left( -cL \cdot R \right) \right) + R.$$

We used the big $O$ notation to suppress any constants, which we may eliminate by paying only $\log(\text{constant})$. Therefore, since the above holds true for all $R$, we may minimize it or choose an $R$ that is satisfactory for our purposes. We may choose $R = O(\log(1/\alpha))$. Therefore,

$$O\left( \frac{1}{\alpha} \exp\left( -cL \cdot R \right) \right) + R = O\left( \log\left( \frac{1}{\alpha} \right) \right),$$

so we conclude the first part.

For the second part we use a similar analysis. To establish the bound, we will bound the corresponding quadratic form of the matrix $\mathbf{Cov}_{x \sim p_\theta^S}(T(x))$. It suffices to bound

$$\left| \mathbf{E}_{x \sim p_\theta^S}\left[ u^T T(x) \cdot T^T(x)u \right] - \mathbf{E}_{x \sim p_\theta}\left[ u^T T(x) \cdot T^T(x)u \right] \right|.$$

To simplify the expression we set $u^T \cdot T(x) = p(x)$ where $p(x)$ is a polynomial of degree $k$.

$$\left| \mathbf{E}_{x \sim p_\theta^S}\left[ u^T T(x) \cdot T^T(x)u \right] - \mathbf{E}_{x \sim p_\theta}\left[ u^T T(x) \cdot T^T(x)u \right] \right|$$

$$= \left| \mathbf{E}_{x \sim p_\theta^S}\left[ p^2 \right] - \mathbf{E}_{x \sim p_\theta}\left[ p^2(x) \right] \right|$$

$$= \left| \mathbf{E}_{x \sim p_\theta}\left[ \left( \frac{\mathbf{1}_S - \alpha}{\alpha} \right) p^2(x) \right] \right|$$

$$\le \left| \mathbf{E}_{x \sim p_\theta}\left[ \left( \frac{\mathbf{1}_S - \alpha}{\alpha} \right) p^2(x) \mathbf{1}_{C_R} \right] + \mathbf{E}_{x \sim p_\theta}\left[ \left( \frac{\mathbf{1}_S - \alpha}{\alpha} \right) R^2 \right] \right|$$

$$\le \frac{1}{\alpha} \mathbf{E}_{x \sim p_\theta}\left[ p^2 \mathbf{1}_{C_R} \right] + R^2$$

The term $\mathbf{E}_{x \sim p_\theta}\left[ p^2(x) \mathbf{1}_{C_R} \right]$ can be bounded using Cauchy-Schwarz as follows

$$\mathbf{E}_{x \sim p_\theta}\left[ p^2(x) \mathbf{1}_{C_R} \right]^2 \le \mathbf{E}_{x \sim p_\theta}\left[ p^4(x) \right] \cdot \mathbf{E}_{x \sim p_\theta}\left[ \mathbf{1}_{C_R} \right] \le C \exp(-cLR),$$

where $C$ is a constant. To obtain the constant $C$, we use Proposition 2.3 (b), which gives

$$\mathbf{E}_{x \sim p_\theta}\left[ p^4 \right] \le (4K_2)^4$$

Substituting back into the original inequality, we get a bound of the form

$$O\left( \frac{1}{\alpha} \exp\left( -\frac{cL}{2R} \right) \right) + R^2.$$

We used the big O notation to suppress any constants, which we may eliminate by paying only $\log(\text{constant})$. Therefore, since the above holds true for all $R$, we may minimize it or choose an $R$ that is satisfactory for our purposes. We may choose $R = O(\log^2(1/\alpha))$. Therefore,

$$O\left( \frac{1}{\alpha} \exp\left( -\frac{cL}{2} \cdot R \right) \right) + R^2 = O\left( \log^2\left( \frac{1}{\alpha} \right) \right),$$

so we conclude.

### B.3 THE PROOF OF THEOREM 1.5

*Proof.* We aim to show that the sequence $\{w^{(t)}\}$ generated by the algorithm converges to an optimal point $w^*$.

We use the non-expansiveness of the projection operator. Namely, the projection operator onto a convex set is non-expansive, meaning for any $x$ and $y$,

$$\|\Pi_{\mathcal{K}}(x) - \Pi_{\mathcal{K}}(y)\| \leq \|x - y\|.$$

Using the convexity and smoothness properties, and the non-expansiveness of the projection operator, we can see the progress achieved after each iteration.

$$\|w^{(t)} - w^*\|^2 \leq \|w^{(t-1)} - h_t g^{(t)} - w^*\|^2.$$

Expanding the right-hand side, we get:

$$\|w^{(t)} - w^*\|^2 \leq \|w^{(t-1)} - w^*\|^2 - 2h_t \langle g^{(t)}, w^{(t-1)} - w^* \rangle + h_t^2 \|g^{(t)}\|^2.$$

Since $f$ is convex, we have:

$$f(w^{(t-1)}) - f(w^*) \leq \langle \nabla f(w^{(t-1)}), w^{(t-1)} - w^* \rangle.$$

Taking expectations conditioned on $w^{(t-1)}$, and noting that $\mathbf{E}\left[g^{(t)} \mid w^{(t-1)}\right] = \nabla f(w^{(t-1)})$, we get:

$$\mathbf{E}\left[f(w^{(t-1)})\right] - f(w^*) \leq \mathbf{E}\left[\langle \nabla f(w^{(t-1)}), w^{(t-1)} - w^* \rangle\right].$$

Combining everything together, taking expectations, and summing over $t = 1$ to $T$,

$$\sum_{t=1}^{T} \mathbf{E}\left[f(w^{(t-1)})\right] - Tf(w^*) \leq \sum_{t=1}^{T} \mathbf{E}\left[\langle \nabla f(w^{(t-1)}), w^{(t-1)} - w^* \rangle\right]$$

$$\leq \sum_{t=1}^{T} \mathbf{E}\left[\langle g^{(t)}, w^{(t-1)} - w^* \rangle\right]$$

$$\leq \frac{1}{2L} \sum_{t=1}^{T} \mathbf{E}\left[\|w^{(t-1)} - w^*\|^2 - \|w^{(t)} - w^*\|^2\right] + h_t^2 \mathbf{E}\left(\|g^{(t)}\|^2\right).$$

Using the boundedness of $\mathbf{E}\left[\|g^{(t)}\|^2\right] \leq b$, and that $h_t = \frac{1}{Lt}$, we have:

$$\sum_{t=1}^{T} h_t^2 \|g^{(t)}\|^2 \leq b \sum_{t=1}^{T} \frac{1}{L^2 t^2} \leq \frac{b}{L^2} \sum_{t=1}^{T} \frac{1}{t^2} \leq \frac{b}{L^2}\left(\frac{\pi^2}{6}\right).$$

So,

$$\frac{1}{T} \sum_{t=1}^{T} \mathbf{E}\left[f(w^{(t-1)})\right] - f(w^*) \leq \frac{\|w^{(0)} - w^*\|^2}{2LT} + \frac{b\pi^2}{12L^2 T}.$$

Finally, from convexity we obtain

$$\mathbf{E}\left[\frac{1}{T} \sum_{t=1}^{T} f(w^{(t-1)})\right] \leq \frac{1}{T} \sum_{t=1}^{T} \mathbf{E}\left[f(w^{(t-1)})\right],$$

hence we conclude. $\qquad\square$

### B.4 THE PROOF OF LEMMA 3.13

Let $v$ denote the output of the rejection sampling procedure to find an unbiased estimate of the gradient. We have

$\mathbf{E}[\|v\|^2]$

$= \underset{z \sim p_\theta^S}{\mathbf{E}} \underset{x \sim p_{\theta^*}^S}{\mathbf{E}} \left[\|T(z) - [T(x)]\|^2\right]$

$= \underset{z \sim p_\theta^S}{\mathbf{E}} \underset{x \sim p_{\theta^*}^S}{\mathbf{E}} \left[\|T(z)\|^2 - 2T(z)^\top T(x) + \|[T(x)]\|^2\right]$

$= \mathrm{Tr}(\mathbf{Cov}[T(z)]) + (\mathbf{E}[\|T(z)\|\|])^2 + \mathrm{Tr}(\mathbf{Cov}[T(x)]) + (\mathbf{E}[\|T(x)\|\|])^2 - 2\langle\mathbf{E}[T(z)], \mathbf{E}[T(x)]\rangle$

$= \mathrm{Tr}(\mathbf{Cov}[T(z)]) + \mathrm{Tr}(\mathbf{Cov}[T(x)]) + \|\mathbf{E}_{x \sim p_{\theta_i}^S}[T(z)] - \mathbf{E}_{x \sim p_{\theta^*}^S}[T(x)]\|^2$

$$\leq k(L + \log\left(\tfrac{1}{\alpha}\right)) + kL + O\left(\lambda^{-1}(L + \log^2\left(\tfrac{1}{\alpha}\right))\right)^2 \log^2\left(\tfrac{1}{\alpha}\right).$$

Since,

$$\|\mathbf{E}_{x \sim p_{\theta_i}^S}[T(z)] - \mathbf{E}_{x \sim p_{\theta^*}^S}[T(x)]\| = \|\nabla \mathcal{L}_S(\theta_i) - \nabla \mathcal{L}_S(\theta^*)\|$$

$$\leq O\left(\left(L + \log^2\left(\tfrac{1}{\alpha}\right)\right)\right)\|\theta_i - \theta^*\|$$

$$\leq O\left(\lambda^{-1}\left(L + \log^2\left(\tfrac{1}{\alpha}\right)\right)\right)\log\left(\tfrac{1}{\alpha}\right)$$

where we used Lemma 3.5 in combination with the assumption of strong convexity of $\mathcal{L}(\theta)$ and the upper bound of the smoothness of $\mathcal{L}_S(\theta)$.

## C   PROOF FOR THEOREM 3.4

To minimize the number of queries to the oracle, our approach cautiously progresses towards the optimal parameter $\theta^*$. Specifically, we incrementally explore increasingly larger convex sets, ensuring each set assigns sufficient probability mass to the set $S$. To this end, for a given threshold $\overline{L}$, we formulate and solve the following constrained optimization problem:

$$\theta^* = \arg\min_\theta \mathcal{L}_S(\theta)$$
$$\text{s.t. } \mathcal{L}(\theta) \leq \overline{L} \tag{6}$$

**Proposition C.1** (Progress of the Truncated Loss along Sublevel Sets). *Suppose $\theta_m^*$ is the minimizer of the optimization problem $\{\min_\theta \mathcal{L}_S(\theta) : s.t. \mathcal{L}(\theta) \leq \overline{L}\}$ for $\overline{L} := L' + m$, where $m \in \mathbb{N}$. Moreover, let $l$ be the smallest integer, such that $\theta^*$ is the solution of the same constrained minimization problem for $\overline{L} := L' + l$. Then, for $m \leq l-1$*

*(a) $\mathcal{L}_S(\theta_m^*) - \mathcal{L}_S(\theta_{m+1}^*) \geq \frac{\mathcal{L}_S(\theta_m^*) - \mathcal{L}_S(\theta^*)}{l-m}$*

*(b) so, particularly, for all $m$, $\mathcal{L}_S(\theta_m^*) - \mathcal{L}_S(\theta_{m+1}^*) \geq \frac{\mathcal{L}_S(\theta_m^*) - \mathcal{L}_S(\theta^*)}{\log\left(\frac{1}{\alpha}\right)}$ .*

**Proof of Proposition C.1**

*Proof.* Suppose $u = \frac{\theta^* - \theta_m^*}{\|\theta^* - \theta_m^*\|_2}$. Then, the functions

$$g(t) = \mathcal{L}_S(\theta_m^* + tu), \quad w(t) = \mathcal{L}(\theta_m^* + tu),$$

are convex. Furthermore, the functions $g$ and $w$ are decreasing and increasing, respectively, for $t \in [0, \|\theta^* - \theta_m^*\|_2]$. Denote by $t_k$ the points such that $w(t_k) = L' + mb + kb$, where $k = 0, \ldots, l-k-1$ and $t_{l-m} = \|\theta^* - \theta_m^*\|_2$. Since the function $w$ is convex and increasing, the length of the line segments $[t_{k-1}, t_k]$ is decreasing. Also, since $g$ is convex and decreasing,

$$g(t_k) - g(t_{k-1}) \leq g(t_{k-1}) - g(t_{k-2}), \quad \text{for } m = 1, \ldots, l.$$

Consequently,

$$\mathcal{L}_S(\theta_m^*) - \mathcal{L}_S(\theta^*) = g(0) - g(\|\theta^* - \theta_m^*\|_2)$$

$$= \sum_{i=1}^{l-m} (g(t_{i-1}) - g(t_i))$$

$$\leq (l-m)(g(t_0) - g(t_1))$$

$$\leq (l-m)(\mathcal{L}_S(\theta_m^*) - \mathcal{L}_S(\theta_{m+1}^*)).$$

Hence, we divide by $l-m$ and obtain the first part. Also, since $l-m \leq \log\left(\tfrac{1}{\alpha}\right)$, we conclude.

$\square$

## C.1 Algorithms Involved in Computing the Output $\hat{\theta}$ with Properties from Theorem 3.4

In this subsection, we include the algorithms used to obtain the output $\hat{\theta}$, which satisfies the properties outlined in the statement of Theorem 3.4. The purpose is twofold: first, to simplify the exposition of the proof of Theorem 3.4; and second, to make the implementation of the procedures more accessible for reproduction.

The first Algorithm 1 generates the desired output, $\hat{\theta}$, while incorporating all subsequent algorithms. The algorithm terminates at line 4 when the successive outputs for Equation (6), corresponding to the sublevel sets $L_1$ and $L_2$ (with $L_2 - L_1 = g$), differ by no more than $\epsilon''$. The parameter $\epsilon''$ is chosen to be small enough such that, at termination, we are within $\epsilon$ of the minimizer of $\mathcal{L}_S(\theta)$. This comparison is justified by Proposition C.1, which compares the successive distances to the distance from the global minimum.

---

**Algorithm 1** Algorithm for Minimizing $\overline{L}$-Constrained Function

1: **for** $\overline{L} \in \{L_{\min} + 1, L_{\min} + 2, \ldots, L_{\min} + (\lceil \log(\frac{1}{\alpha}) \rceil + 2)\}$ **do**
2:     Execute projected stochastic gradient descent Algorithm 2 {For sublevel set $\overline{L}$}
3:     Take the output $\theta_{\bar{L}}$ of Algorithm 2 and calculate $\mathcal{L}_S(\theta_{\bar{L}})$
4:     **if** $|\mathcal{L}_S(\theta_{\bar{L}-1}) - \mathcal{L}_S(\theta_{\bar{L}})| \leq \epsilon''$ **then**
5:         Stop and return $\theta_{\bar{L}}$
6:     **end if**
7: **end for**

---

Next, we have the projected stochastic gradient descent (PSGD) algorithm that is performed at each sublevel set.

---

**Algorithm 2** Projected SGD Algorithm Given Truncated Samples and Sublevel Set

**Require:** $\bar{L}$, $\{x_i\}_{i=1}^n$, each $x_i \sim p_{\theta^*}^S$, Initial $\theta_0 \in \mathbb{R}^k$, where $\theta_0 = \frac{1}{n}\sum_{i=1}^n T(x_i)$, step size parameter $L$
1: **for** $t = 1, \ldots, N$ **do**
2:     Sample $x^{(t)}$ from the data distribution
3:     $h_t \leftarrow \frac{1}{Lt}$
4:     $g^{(t)} \leftarrow$ Sample Gradient$(\theta_t, x^{(t)})$
5:     $\theta_{t+1} \leftarrow \theta_t - h_t g^{(t)}$
6:     Project $\theta_{t+1}$ onto $\{\theta : \mathcal{L}(\theta) \leq \bar{L}\}$
7: **end for**
8: **return** $\theta_N$

---

Finally, we have the process where rejection sampling is performed.

---

**Algorithm 3** Sample Gradient

**Require:** $x, \theta$
1: **while** True **do**
2:     Sample $z \sim p_\theta$
3:     **if** $M_S(z) = 1$ where $M_S$ is the membership oracle **then**
4:         **return** $T(z) - T(x)$
5:     **end if**
6: **end while**

---

## C.2 The proof of Theorem 3.4

*Proof.* The parameter $\theta$ that will satisfy the properties of the theorem will be the output of the Algorithm 1. However, before handling this, we first show that there is a way to amplify the probability

of the output $\theta_{\bar{L}}$ as it appears on line 3, by running the PSGD independently and then choosing the best output among all the trials.

Apply Theorem 3.12 (PSGD) to the optimization problem in Equation (6) for the sublevel sets $\bar{L} = L_{\min} + m + \epsilon_S$, where $m \in [0, \log\left(\frac{1}{\alpha}\right) + 2]$, $L_{\min} = \mathcal{L}\left(\mathbf{E}_{x \sim p_{\theta*}^S}[T(x)]\right)$, $\theta_0 = \frac{1}{N}\sum_{i=1}^N T(x_i)$, and $\|\theta_0 - \mathbf{E}_{x \sim p_{\theta*}^S}[T(x)]\|_2 \leq \epsilon_S$.

Choose $\epsilon_S = \frac{\epsilon}{L}$, which holds for $N \geq O\left(\frac{kL^3}{\epsilon^2}\log\left(\frac{1}{\delta}\right)\log^2\left(\frac{1}{\alpha}\right)\right)$. Given the $L$-smoothness of $\mathcal{L}$ and the fact that $\nabla\mathcal{L}(\mathbf{E}_{x \sim p_{\theta*}^S}[T(x)]) = 0$, we obtain the bound $\|\nabla\mathcal{L}(\theta)\| \leq L \cdot r$ for all $\theta \in \{\theta : \|\theta - \mathbf{E}_{x \sim p_{\theta*}^S}[T(x)]\| \leq r\}$.

Therefore, with this choice of $\epsilon_S$, we have $\mathcal{L}(\theta_0) - L_{\min} \leq \epsilon$, and thus the following bound holds:

$$\|w^{(0)} - w^*\|^2 = \|\theta_0 - \theta^*\|^2 \leq \frac{2}{\lambda}\left(\mathcal{L}(\theta^*) - \mathcal{L}(\theta_0)\right) \leq O\left(\frac{1}{\lambda}\log\left(\frac{1}{\alpha}\right)\right).$$

Hence, Theorem 3.12 in our setup gives an upper bound:

$$\frac{C\log^2(1/\alpha)}{2(L + \log(1/\alpha))T} + \frac{b\pi^2}{12(L + \log(1/\alpha))^2 T}$$

So, for:

$$T \geq \frac{1}{\epsilon}\left(\frac{C\log^2(1/\alpha) + 6(L + \log(1/\alpha))b\pi^2}{12(L + \log(1/\alpha))^2}\right)$$

Theorem 3.12 gives us a $\bar{\theta}$ such that:

$$\mathbf{E}\left[f(\bar{\theta})\right] - f(\theta^*) \leq \epsilon.$$

From Markov's inequality, we get:

$$\mathbf{Pr}\left[f(\bar{\theta}) - f(\theta^*) \geq 3\epsilon\right] \leq \frac{1}{3}$$

We can easily amplify this probability by repeating this process independently and hence obtaining a sequence of $\bar{w}_1, \bar{w}_2, \ldots, \bar{w}_m$, and then choosing:

$$\bar{w} = \operatorname*{argmin}_{\bar{w}_i} f(\bar{w}_i).$$

Since $\bar{\theta} := \bar{w}$ satisfies:

$$\mathbf{Pr}\left[f(\bar{\theta}) - f(\theta^*) \geq 3\epsilon\right] \leq \left(\frac{1}{3}\right)^m,$$

by choosing $m \geq \log(\delta)/\log(1/3)$, we obtain an $\bar{w}$ that satisfies:

$$\mathbf{Pr}\left[f(\bar{\theta}) - f(\theta^*) \geq 3\epsilon\right] \leq \delta.$$

To accomplish this, however, we need access to the value of:

$$\mathcal{L}_S(\bar{w}_i) = \mathcal{L}(\bar{w}_i) + \log\left(\mathbf{Pr}_{x \sim p_{\bar{w}_i}}[x \in S]\right).$$

Since we have access to $S$ only through its oracle, in order to calculate $\mathbf{Pr}_{x \sim p_{\bar{w}_i}}[x \in S]$, we use concentration for a Bernoulli random variable. More specifically, by Hoeffding's inequality, we need $O\left(\frac{1}{\epsilon}\log\left(\frac{1}{\alpha}\right)\right)$ samples to estimate $\log\left(\mathbf{Pr}_{x \sim p_{\bar{w}_i}}[x \in S]\right)$ $\epsilon$-close with probability at least $1 - \delta$.

Therefore, for each sublevel set $L_{\min} + m + \epsilon_S$, when we perform PSGD, we have an output $\theta_m$ such that, with high probability, it achieves high precision:

$$\mathcal{L}_S(\theta_m) - \mathcal{L}_S(\theta_m^*) < \epsilon'$$

where $\theta_m^*$ is the solution to the optimization problem Equation (6) for the sublevel set $L_{\min} + m + \epsilon_S$.

Suppose Algorithm 1 is terminated when the if statement on line 4 is verified. Then, the $\theta_{\bar{L}}$ generated on line 3 satisfies $\left|\mathcal{L}_S(\theta_{\bar{L}-1}) - \mathcal{L}_S(\theta_{\bar{L}})\right| \leq \epsilon''$. Fix $\epsilon > 0$, and set $\epsilon'' = \frac{\epsilon}{\log(1/\alpha)}$. We claim that for this choice of $\epsilon''$, Algorithm 1 gives output $\theta_f$ with high probability, such that $\mathcal{L}_S(\theta_f) - \mathcal{L}_S(\theta^*) \leq \epsilon$.

Observe that Algorithm 1 terminates. By Corollary 3.10, and we have

$$\theta^* \in \left\{ \mathcal{L}(\theta) - L_{\min} \leq \log\left(\frac{1}{\alpha}\right) \right\} \subseteq \left\{ \mathcal{L}(\theta) - \mathcal{L}(\theta_0) \leq \log\left(\frac{1}{\alpha}\right) \right\}$$

Consequently, there are at least two distinct sublevel sets within the range of Algorithm 1 that contain $\theta^*$. Therefore, for each of these optimization problems, if we run PSGD with an accuracy of $\epsilon' = \frac{\epsilon''}{2}$, the stopping criterion on line 4 of Algorithm 1 will be triggered, as both sublevel sets approximate the same minimizer with high probability and accuracy $\frac{\epsilon''}{2}$.

Now that we have established that Algorithm 1 terminates we distinguish two cases, namely $\theta^*$ is either in $\{\theta : \mathcal{L}(\theta) - \mathcal{L}(\theta_0) \leq m\}$ or it is not. Suppose the first case, i.e. $\theta^* \in \{\theta : \mathcal{L}(\theta) - \mathcal{L}(\theta_0) \leq m\}$. Since $\theta^*$ is the minimizer of the convex function $\mathcal{L}_S(\theta)$, and it is inside the optimization domain $\{\theta : \mathcal{L}(\theta) - \mathcal{L}(\theta_0) \leq m\}$ it implies that $\theta_m$ the output of the PSGD satisfies $\mathcal{L}_S(\theta_m) - \mathcal{L}_S(\theta^*) \leq \epsilon'$.

We deal now with the other case, i.e. when $\theta^* \notin \{\theta : \mathcal{L}(\theta) - \mathcal{L}(\theta_0) \leq m\}$. From Proposition C.1

$$\mathcal{L}_S(\theta_m^*) - \mathcal{L}_S(\theta_{m+1}^*) \geq \frac{\mathcal{L}_S(\theta_m^*) - \mathcal{L}_S(\theta^*)}{\log\left(\frac{1}{\alpha}\right)}$$

And suppose we get $\theta_f = \theta_{m+1}$. From our assumption, recall that

$$|\mathcal{L}_S(\theta_{m+1}) - \mathcal{L}_S(\theta_m)| \leq \epsilon'',$$

where $\epsilon''$ is our threshold value for stopping Algorithm 1.

$$\mathcal{L}_S(\theta_m^*) - \mathcal{L}_S(\theta_{m+1}^*) \leq |\mathcal{L}_S(\theta_m) - \mathcal{L}_S(\theta_{m+1})| + \frac{2\epsilon}{\log(1/\alpha)}$$

$$\leq \epsilon'' + \frac{2\epsilon}{\log(1/\alpha)}$$

Therefore, by our choice of $\epsilon''$, $\epsilon'' = \frac{\epsilon}{\log(1/\alpha)}$, we get that

$$\frac{\mathcal{L}_S(\theta_m^*) - \mathcal{L}_S(\theta^*)}{\log\left(\frac{1}{\alpha}\right)} \leq \frac{3\epsilon}{\log(1/\alpha)}.$$

$$\mathcal{L}_S(\theta_f) - \mathcal{L}_S(\theta^*) \leq \mathcal{L}_S(\theta_m^*) - \mathcal{L}_S(\theta^*) \leq 3\epsilon.$$

In the previous case, when termination, of Algorithm 1, occurred at line 4, we obtained a suitable output by virtue of identifying a threshold value $\epsilon''$ so small that if the difference in the values of the approximated minima between two successive sets was bounded by $\epsilon''$, then even if we had continued our search, our progress in terms of reducing the value of $\mathcal{L}_S$ would have been negligible.

Finally, it remains to show that throughout the process, we always maintain a linear mass in terms of $\alpha$. This follows directly, since upon termination of Algorithm 1, the output parameter $\theta$ lies in the sublevel $\bar{L} = \mathcal{L}(\theta_0) + m$, which satisfies $\bar{L} \leq \mathcal{L}(\theta^*) + \epsilon_S + 2$. Therefore, by Observation 3.8, we conclude that all parameters $\theta'$ accessed by Algorithm 1 possess at least $\alpha e^{-\epsilon_S - 2}$ mass.

$\square$

