# OpenReview forum: "Oracle efficient truncated statistics"
_ICLR.cc/2025/Conference — ICLR 2025 Poster_

### Official Review · Reviewer_JmS1 · 2024-10-23

**Soundness:** 3
**Presentation:** 2
**Contribution:** 3
**Rating:** 6
**Confidence:** 3

**Summary:**

This paper studies the problem of learning an exponential family with truncated samples. Specifically, for any distribution $p$, the truncated distribution $p^S$ corresponds to the distribution of $p$ restricted to a set $S$. The paper shows that if a distribution $p$ is selected from an exponential family, and we are able to observe samples from $p^S$ and access a membership oracle for $1\\{x \in S\\}$, then one can find an estimator $\hat{p}$ such that $KL(p^S ||\hat{p}^S) \leq \epsilon$. The main technical contribution is to obtain bounds on the sample complexity and oracle complexity that depend poly-logarithmically on $\alpha:=p^S[S]$, with a computationally efficient estimator $\hat{p}$.

**Strengths:**

The main strength of this paper comes from the polylogarithmic dependency on the parameter $\alpha$ in both the sample and oracle complexities. As far as I'm aware, this seems to be original, at least for bounding the KL-divergence of the truncated distributions themselves. The paper also introduces several new techniques, such as the minimization of the "truncated" negative log-likelihood (NLL) constrained by a bound on the original NLL.

**Weaknesses:**

The main weakness of the paper is its presentation; it is quite difficult to follow the proofs provided in Section 3. I have the following specific comments:

1. The paper emphasizes that its primary contribution is improving the poly($1/\alpha$) dependency from Lee et al. (2023) to poly-log($1/\alpha$). However, a quick look at that paper reveals that Lee et al. (2023) focuses on estimating the parameter, not the truncated distribution. Therefore, I am unsure how significant this improvement is.

2. Section 3 is quite dense, and the logical flow of the proof is not well explained. Overall, the section feels extremely unstructured.

I suggest that the authors improve the writing in Section 3 by, for example, providing a clear roadmap of the proof at the beginning of this section and clearly explaining how each lemma contributes to the final proofs. Please also refer to the "Questions" section.

**Questions:**

1. I assume you are using $\theta^*$ to denote the ground truth parameter throughout the paper, but in Corollary 3.9, it seems to be defined differently. Is the $\theta^*$ in Corollary 3.9 the ground truth as well?

2. In the second assertion of Corollary 3.10, why is the consequence stated for $\theta^*$ when your condition is for $\theta^0$?

3. Can you provide a detailed proof of Corollary 3.10? Specifically, it is unclear how the definition $\theta^0 = \mathbb{E}_{p}[X]$ plays a role in the proof.

4. In the proof of Theorem 3.3, how do you derive $p_{\theta}(S) \ge \frac{1}{\alpha^2} e^{-\epsilon}$? I assume you are referring to Corollary 3.10, not Corollary 3.9? Moreover, what is the significance of this result in the overall proof?

5. Can you clarify the terms $w$ and $f$ in the proof of Theorem 3.3? How do they relate to $\theta$ and $\mathcal{L}$?

6. Typos:
    - Line 466: $\min\\{\min...\ \\}$
    - Line 519: It should be "Here" instead of "Where" to start a new sentence.

---

> ### Author Response · Authors · 2024-11-20
> **Thank you for the review. We have implemented the roadmap to the proof, along with the other minor points you suggested.**
>
> The paper emphasizes that its primary contribution is improving the poly(1/α) dependency from Lee et al. (2023) to poly-log(1/α). However, a quick look at that paper reveals that Lee et al. (2023) focuses on estimating the parameter, not the truncated distribution. Therefore, I am unsure how significant this improvement is.
>
> $\textbf{Response:}$ Our result implies the parameter estimation of Lee et al. (2023) (see Corollary 1.7, line 156), as their assumptions include the strong convexity of $\mathcal{L}_S$ (or properties that imply strong convexity). Consequently, close values in $\mathcal{L}_S$ (as per our result) immediately imply close values in $\theta$ (see Corollary 1.7). However, in our setting, we do not assume strong convexity, allowing us to handle a more general class of problems, albeit with a trade-off in guarantees.
>
>
> Section 3 is quite dense, and the logical flow of the proof is not well explained. Overall, the section feels extremely unstructured.
> I suggest that the authors improve the writing in Section 3 by, for example, providing a clear roadmap of the proof at the beginning of this section and clearly explaining how each lemma contributes to the final proofs. Please also refer to the "Questions" section.
>
> $\textbf{Response:}$ Thank you for the suggestion. Taking this into account, we have included a roadmap starting on line 365.
>
> Questions:
> I assume you are using θ∗ to denote the ground truth parameter throughout the paper, but in Corollary 3.9, it seems to be defined differently. Is the θ∗ in Corollary 3.9 the ground truth as well? In the second assertion of Corollary 3.10, why is the consequence stated for θ∗ when your condition is for θ0?
>
> $\textbf{Response:}$ You are absolutely right; it was a typo. The stated consequence applies to $\theta^0$, i.e., $p_{\theta}(S) \geq p_{\theta^0} \alpha \exp(-\epsilon)$.
>
> Can you provide a detailed proof of Corollary 3.10? Specifically, it is unclear how the definition θ0=Ep[X] plays a role in the proof.
>
> $\textbf{Response:}$ Certainly, here is a detailed proof of Corollary 3.10. First, I hope any misunderstanding wasn’t due to the typo $\theta^0 =E_p(T(x))$. Now, to proceed with the proof:
> From Corollary 3.9, we have that $p_{\theta^0}(S) > \alpha$, as $\theta^0$ is the parameter that minimizes $\mathcal{L}(\theta)$ and thus also minimizes $\mathcal L_S(\theta)$. Consequently, $\theta^0$ is the only feasible parameter for this sublevel set due to the strong convexity of $\mathcal L(\theta)$. Therefore, we can invoke part 2 of Corollary 3.9. Specifically, applying Corollary 3.9, part 2 (exponential decrease) with $\theta_2 = \theta^\ast$ and $\theta_1 = \theta^0$, we obtain: $P_{\theta_2}(S) \leq P_{\theta_1}(S) \exp(\mathcal L(\theta_1) - \mathcal L(\theta_2)).$
> Since $P_{\theta_2}(S) = \alpha$ and $P_{\theta_1}(S) \leq 1$, taking the logarithm and rearranging terms yield: $\mathcal L(\theta^\ast) \leq \mathcal L(\theta^0) + \log(1/\alpha).$
> The second part follows immediately from Observation 3.8, applied to $\mathcal L(\theta) \leq \mathcal L(\theta^0) + \log(1/\alpha) + \epsilon$ and the minimizer $\theta^0$.
>
> In the proof of Theorem 3.3, how do you derive pθ(S)≥1α2e−ϵ? I assume you are referring to Corollary 3.10, not Corollary 3.9? Moreover, what is the significance of this result in the overall proof?
>
> $\textbf{Response:}$ Correct, this is Corollary 3.10. The significance of this result is that, for every $\theta$ in the projection domain $D$ (line 515), the probability mass that each $\theta$ assigns to $S$, i.e., $P_{x \sim p_{\theta}}[x \in S]$, is at least a constant fraction of $\alpha^2$. This clarification has been added to the roadmap to the proof on lines 365–377.
>
>
>
>
> Can you clarify the terms w and f in the proof of Theorem 3.3? How do they relate to θ and L?
>
> $\textbf{Response:}$ Thank you this has been clarified, $f$ is $\mathcal L_S$ and $w$ is $\theta$.
> Typos: Line 466: min{min... }
> Line 519: It should be "Here" instead of "Where" to start a new sentence.
>
> $\textbf{Response:}$ Thank you, we fixed these typos/mistakes.

---

### Official Review · Reviewer_uTFY · 2024-10-27

**Soundness:** 4
**Presentation:** 3
**Contribution:** 3
**Rating:** 8
**Confidence:** 3

**Summary:**

This paper studies the problem of learning a high-dimensional probability distribution from "truncated samples" -- that is, the goal is to learn a distribution $p$ given samples from $p$ conditioned on some event $S$, called the "survival set".This is not possible with reasonable (polynomial) sample complexity for an arbitrary distribution $p$, so, like prior work, the paper focuses on exponential family distributions.

More concretely, the model is that one gets:

-- samples from $p$ conditioned on $S$; the set $S$ is not known to the algorithm
-- access to an oracle which takes as input a probability distribution $q$ and returns a samples from $q$ conditioned on $S$

Here $p$ and $q$ should be exponential family distributions so that they have a short description which can be handed to the oracle. There is a further constraint that the probability mass of $S$ under $p$ and $q$ is at least some parameter $\alpha > 0$. As $\alpha$ gets small, this problem gets harder, since we are observing only samples from some low-probability event under $p$.

Prior work showed how to accomplish this task for exponential family distributions, with sample complexity and running time depending polynomially on dimension of the distribution and the inverse of the accuracy to which the distribution is learned, but super-polynomially on $1/\alpha$. (Prior work achieved only quasipolynomial dependence.)

This dependence on survival probability is important for overall running time, because one way to implement the sampling oracle is via rejection sampling, using a membership oracle for $S$ and a sampling algorithm for $q$. But the running time of such an implementation will be about $1/\alpha$ to get a single sample.

The main contribution of this work is to give an algorithm with running time and sample complexity having polynomial dependence on dimension, inverse of the accuracy, and $1/\alpha$. My understanding is that this result is novel even in the special case of Gaussian distributions.

The problem is extremely well motivated -- truncation is a common situation for "in the wild" datasets. And the technical contribution is certainly sufficient for ICLR. I recommend acceptance.

**Strengths:**

See above

**Weaknesses:**

See above

**Questions:**

Minor comments:

-- It would be nice if the analogues of 1.6 and 1.7 from prior work were stated formally rather than alluded to in prose, for easier comparison. For example, I am still a little unsure after reading the prose if prior work gives an algorithm with $poly(1/\alpha)$ dependence for the special case of Gaussians?

---

> ### Author Response · Authors · 2024-11-20
> **Thank you for the review. It was a very good insight to make the comparison with previous literature easier to compare.**
>
> -- It would be nice if the analogues of 1.6 and 1.7 from prior work were stated formally rather than alluded to in prose, for easier comparison. For example, I am still a little unsure after reading the prose if prior work gives an algorithm with poly(1/α) dependence for the special case of Gaussians?
>
>
> $\textbf{Response:}$ Thank you for the suggestion on improving our exposition. To address this, we have included a section in the appendix rephrasing these results to reflect our perspective. Interestingly, the exponential family algorithm by Lee et al. represents an improvement over the pre-existing result for Gaussians, as their abstraction streamlined the proofs and avoided relying on inefficient bounds for how the mass assigned to a truncation set $S$ changes with parameter adjustments. Our result is tight in this sense, providing an optimal solution for the Gaussian case as well, particularly in identifying high-mass regions.

---

### Official Review · Reviewer_3xik · 2024-11-01

**Soundness:** 1
**Presentation:** 1
**Contribution:** 3
**Rating:** 3
**Confidence:** 4

**Summary:**

The authors address the problem of estimating a parametric distribution from truncated samples.
They propose an efficient procedure for parameter estimation within an exponential family under these conditions, improving the dependence on the probability mass of the survival set from a super-polynomial to a polynomial rate.

**Strengths:**

This work fits well within the existing literature; the efficiency problem is well-motivated and clearly defined.
The authors present a clear improvement in terms of computational efficiency.

**Weaknesses:**

The presentation of the paper is poor:
- The structure of the paper is unclear, the main approach is not even presented in the main body of the paper;
- There are significant redundancies; for example, Section 2.1 is nearly identical to Section 1.1;
- There are many typos.

The proof of the main result contains several gaps (see Questions), and I do not find it convincing at this stage.

There are no numerical experiments to illustrate their theoretical results or to compare with existing approaches.

**Questions:**

- l.45: the definition of $A(\theta)$ is wrong given the previous definition of $p_\theta$;

- Definition 1.1 should not be a definition but a proposition;

- Theorem 3.3 and Theorem 3.4 are nearly identical, differing only in their dependencies on $\alpha$ and $\alpha^2$. I find it confusing to present both; what is the rationale behind this choice?;

- In Lemma 3.7, $u$ and $\hat{u}$ are never defined, I guess that they authors meant $\theta^*_S$ and $\hat{\theta}$?


**Proof of Theorem 3.3:**

- Inconsistent notation for $\theta_0$ that appears as $\theta_0$ and $\theta^0$;

- l.519: why is there a $L^3$ term in the lower bound on $N$? It appears as $L$ in Lemma 3.7;

- l.521: the bound on $\lVert \theta_0 - \mathbb{E}[T(x)] \rVert_2$ holds with probability at least $1-\delta$ (according to Lemma 3.7). However, the bound is treated as if it hold almost surely in the rest of the proof. I believe that this will cause an issue when bounding the deviation of $f(\bar{\theta})$ from $f(\theta^*)$ (l.529). Can it be easily fixed?

- I do not understand the inequalities l.525. Since it it never defined, I assumed that $w^{(0)} \coloneqq \theta_0$. In this case I don't understand where the $1/\lambda$ multiplicative term comes from in the first inequality. Moreover, I cannot reproduce the second inequality using the current information in the proof. Can the authors clarify this part?

- There are many typos in the proof: $f$ instead of $\mathcal{L}$, $w$ is not defined, l.525 the bound holds for the norm of the gradient, the index $\theta$ is not defined l.524...

---

> ### Author Response · Authors · 2024-11-20
> **Thank you for the review. Implementing and adjusting the aspects you brought to our attention has significantly improved our work.**
>
> Questions:
> l.45: the definition of A(θ) is wrong given the previous definition of pθ;
>
> $\textbf{Response:}$ Thank you, we fixed it.
>
> Definition 1.1 should not be a definition but a proposition;
>
> $\textbf{Response:}$ This is a definition; we define $\mathcal{L}$ as a function possessing specific properties.
>
> Theorem 3.3 and Theorem 3.4 are nearly identical, differing only in their dependencies on α and α^2, I find it confusing to present both; what is the rationale behind this choice?;
>
> $\textbf{Response:}$ The reason is that the theorem with dependency on $\alpha$ is sharp (line 374-5), so in that sense it gives a complete answer to how low can the dependency on $\alpha$ in terms of finding a suitable optimization domain that has at least a constant fraction of mass of $\alpha$.
> However its proof is a little bit tedious and technical, but as outlined in lines 376-377 it is just a repeated application of the techniques used in theorem 3.3.
> To summarize the proof for $\alpha^2$ is much cleaner and demonstrates the main idea of our paper, without tiring the reader. On the other hand, the inclusion of theorem 3.4 gives best possible dependency of alpha possible.
>
> In Lemma 3.7, u and u^ are never defined, I guess that they authors meant θS∗ and θ^?
>
> $\textbf{Response:}$ Thank you that was a typo, it is as you suspected. It has been fixed.
>
> Proof of Theorem 3.3:
> Inconsistent notation for Θ0 that appears as $Θ0 and θ0; l.519:
>
> $\textbf{Response:}$ The notation is not inconsistent; we use $\theta^0$ as it appears on corollary 3.10. This has been made clear in the newest version, thank you.
>
> why is there a L^3 term in the lower bound on N?
>
> $\textbf{Response:}$ Notice that the bound there is not $\epsilon$ but $\epsilon/L$, therefore the $L^3$ term. I hope this has made it clear.
>
> It appears as L in Lemma 3.7; l.521: the bound on ‖θ0−E[T(x)]‖2 holds with probability at least 1−δ  (according to Lemma 3.7). However, the bound is treated as if it hold almost surely in the rest of the proof. I believe that this will cause an issue when bounding the deviation of f(θ¯) from f(θ∗(l.529). Can it be easily fixed?
>
> $\textbf{Response:}$ Everything we write subsequently, in the proof, holds almost surely on that event. And then the result follows from a union bound, this is a common argument in this literature, i.e. finding a good initialization point with good probability and then conditional on that amplifying the probability of the output of gradient descent. If this doesn’t clarify things please inquire further.
>
> I do not understand the inequalities l.525. Since it it never defined, I assumed that w(0):=θ0. In this case I don't understand where the 1/λ multiplicative term comes from in the first inequality.
>
> $\textbf{Response:}$ Indeed $w^{(0)} = \theta_0$, the $1/\lambda$ should have been after the $\leq $ sign. Thank you for catching that. We have also added that w, f come from the statement of the theorem 3.12.
>
> Moreover, I cannot reproduce the second inequality using the current information in the proof. Can the authors clarify this part?
>
> $\textbf{Response:}$ Is the inequality $| \theta_0 - \theta^\ast|^2 / \lambda \leq O( \log^2 ( 1/\alpha) )$ that you are inquiring for? In that case it is from strong convex i.e. $| \theta_0 - \theta^\ast| ^2  \leq 2/\lambda \left (\mathcal L ( \theta_0) - \mathcal L( \theta^\ast) \right ).$
>
> There are many typos in the proof: f instead of L, w is not defined, l.525 the bound holds for the norm of the gradient, the index θ is not defined l.524.
>
> $\textbf{Response:}$ The variables $f$ and $w$ were introduced in the statement of Theorem 3.12 (PSGD) to aid the exposition, but this was not explicitly stated. We have corrected the typo regarding the norm of the gradient and revised the text to clarify this connection, thereby improving the overall clarity of the exposition.

---

> > ### Comment · Reviewer_3xik · 2024-11-25
> > **Reply**
> >
> > Thank you for your detailed reply, I agree with most of the points.
> >
> > However one crucial issue has not been addressed. My initial comment was
> > - l.521: the bound on $\lVert \theta_0 - \mathbb{E}[T(x)] \rVert_2$ holds with probability at least $1-\delta$ (according to Lemma 3.7). However, the bound is treated as if it hold almost surely in the rest of the proof. I believe that this will cause an issue when bounding the deviation of $f(\bar{\theta})$ from $f(\theta^*)$ (l.529). Can it be easily fixed?
> >
> > I understand the union bound argument but it seems to me that it is not applied on the event that I was referring to.
> >
> > Assuming that $N$ is large enough (with a dependence on a fixed $\delta$), there exists an event of probability at least $1-\delta$ such that $\lVert \theta_0 - \mathbb{E}[T(x)] \rVert_2$. After some steps, the authors show that, on this event, $\mathbb{P}(f(\bar{\theta}) - f(\theta^*) \geq 3 \varepsilon) \leq 1/3$. The authors claim that they can "can easily amplify this probability by repeating this process independently" to find a $\hat{\theta}$ such that $\mathbb{P}(f(\bar{\theta}) - f(\theta^*) \geq 3 \varepsilon) \leq (1/3)^m$. Is the process repeated on the same event of probability at least $1-\delta$? Where does the $1-\delta$ mass of this high-probability event appear in the proof? It seemed to me that it simply disappears.

---

> > > ### Author Response · Authors · 2024-11-26
> > >
> > > Thank you for your thoughtful review and for engaging actively in the discussion of our work. We appreciate the opportunity to clarify and refine our arguments.
> > >
> > > To address your concerns, let us outline the process in detail:
> > >
> > > 1. We begin by fixing an $N(\delta)$ sufficiently large such that the event $\| \theta_0 - \mathbb{E}[T(x)] \|_2 \leq \epsilon$ holds with probability at least $1 - \delta$.
> > >
> > > 2. On this high-probability event (denoted as $\mathcal{E}$), we treat $\theta_0$ as fixed and perform $m$ independent runs of PSGD. Each run produces an output $\hat{\theta}$, and we ensure that, with probability at least $1 - \delta$ (conditional on $\mathcal{E}$), $f(\hat{\theta}) - f(\theta^\ast) < \epsilon$.
> > >
> > > 3. By taking the conditional probability of the desired outcome on $\mathcal{E}$ and then taking the expectation over $\mathcal{E}$, we obtain that the total probability of success is at least $(1 - \delta)(1 - \delta) = 1 - 2\delta + \delta^2$.
> > >
> > > 4. Importantly, we note that the conditional probability of success given $\mathcal{E}$ is always at least $1 - \delta$, which ensures that the amplification process is consistent.
> > >
> > > To directly address your question about the union bound and the treatment of the high-probability event:
> > >
> > > The event $\mathcal{E}$ corresponding to the initialization is treated explicitly in our argument. We fix $\delta$ first, then ensure that $N$ is large enough to achieve the desired probability. Subsequently, we choose $m$ to at least match this $\delta$, ensuring that the process of amplification operates on this high-probability event. Since we can make the probability of $\mathcal{E}$ arbitrarily close to $1$ by taking $N$ large enough, the residual probability $\delta$ suffices for our purposes.
> > >
> > > Does this explanation resolve your concern? Specifically, we aimed to clarify how the initial high-probability event is used and why we did not emphasize the probability of good initialization beyond ensuring that it is sufficiently high for our analysis. If further clarification is needed, we'd be happy to elaborate.

---

### Official Review · Reviewer_7NRY · 2024-11-07

**Soundness:** 3
**Presentation:** 2
**Contribution:** 3
**Rating:** 8
**Confidence:** 4

**Summary:**

This paper revisits the problem of learning an exponential family of distributions under truncated samples. The main focus is the dependence of the runtime and sample complexity on the mass of the truncation set.

In the truncated samples model, the samples are only observed when they fall into a subset S of the domain that has some measure $\alpha$. Such a sampling oracle can be implemented using a membership oracle for the set S. Although for constant values of the mass $\alpha$ the sample (and query) complexity asymptotically matches the bounds from prior work on the problem, there is an exponential improvement in the dependence on $1/\alpha$, when $\alpha$ gets close to $0$. Another difference from prior work is the use of weaker assumptions, which also leads to a weaker type of learner. More specifically, while prior work by Lee et’ al can estimate the parameter vector of the exponential family up to error $\varepsilon$, in this paper, the algorithm is a proper learner that outputs an exponential family that has small KL divergence with the true distribution.

From a technical standpoint, the novelty lies in the definition of a sequence of constrained optimization problems. Specifically, the authors use projected SGD in a region where the negative log likelihood is not allowed to exceed a certain threshold during each iteration. This prohibits the algorithm from moving to regions in the parameter space that would assign very small mass to the truncation set and seems to be crucial for achieving the exponential improvement in terms of the inverse mass $1/\alpha$ of that set S.



Minor comments:

-Line 53: “resoursed”->“resourses”

-Lines 109-114: The example here needs some rephrasing. I get the point you are making, but technically, when you fix a set (S=[0,1] in this case), the mass assigned to it and the variance of the distribution, there is a unique distribution (NOT multiple ones) satisfying these constraints. Maybe you could say $O(\alpha)$ mass.

-Lines 131 and 371: In the second argument of the KL distance, it should be $\hat{\theta}$ in the subscript.

-Line 388: Usually the word “requires” is used for lower bounds. Maybe say “a good initialization can be found”

-Line 399: “estimate of”->”bound on”

-Line 448: Proof of Observation 3.8: The statement seems reasonable,  but I think that the proof is incomplete. You show that the exponential is at most 1, but you need something stronger since it could be that $Pr_{\theta^\prime}[S]>Pr_{\theta}[S]$, right?

-Line 490: “$SE(K,\beta)$”->“$SE(K^2,\beta)$”

**Strengths:**

The paper gives an interesting perspective to an important question in statistics involving data corruption and potentially expands the applicability of state-of-the-art techniques to settings where the corruption is much stronger.

**Weaknesses:**

I believe the authors should make the assumptions they need to use clearer from the introduction to improve the presentation quality. For example, the strong convexity and smoothness assumption is not mentioned in the statement of Theorem 1.5, which makes it seem too strong.

**Questions:**

In section 1.1 (line 133), you mention that: “the fact that our algorithm only requires an α-truncated sample oracle is what enables us to achieve a polynomial number of oracle calls to the membership oracle to S”. Can you elaborate on that? Is this not due to the threshold that you introduce in the optimization problems? It seems that assuming the truncated oracle access only allows logarithmic dependence instead of polynomial, which you would get via rejection sampling.

---

> ### Author Response · Authors · 2024-11-20
> **Thank you for the review and for helping us improve our work.**
>
> Minor comments:
> -Line 53: “resoursed”->“resourses”
>
> $\textbf{Response:}$ Thank you we fixed it.
>
> -Lines 109-114: The example here needs some rephrasing. I get the point you are making, but technically, when you fix a set (S=[0,1] in this case), the mass assigned to it and the variance of the distribution, there is a unique distribution (NOT multiple ones) satisfying these constraints. Maybe you could say O(α) mass.
>
> $\textbf{Response:}$ In the example, we considered the family $N(\mu, \sigma^2)$ without fixing the variance. However, we fixed the variance in the case where the learner can make queries to ensure that the probability bound holds. We have added a clarification to the example to make this more explicit. Thank you for your feedback.
>
>
> -Lines 131 and 371: In the second argument of the KL distance, it should be
> θ^ in the subscript.
> -Line 388: Usually the word “requires” is used for lower bounds. Maybe say “a good initialization can be found”
> -Line 399: “estimate of”->”bound on”
>
> $\textbf{Response:}$ Thank you for the suggestions. We have implemented all of them.
>
> -Line 448: Proof of Observation 3.8: The statement seems reasonable, but I think that the proof is incomplete. You show that the exponential is at most 1, but you need something stronger since it could be that Prθ′[S]>Prθ[S] right?
>
> $\textbf{Response:}$ For the proof of Observation 3.8: Since $\theta'$ is the minimizer of the constrained optimization problem $\bar{L}$, we immediately deduce that $\mathcal L_S(\theta') \leq \mathcal L_S(\theta)$. Interestingly, this result is purely a consequence of this observation. Specifically, by rearranging terms, we obtain: $
> \log(\Pr_{\theta'}[S]) \leq \log(\Pr_{\theta}[S]) + \mathcal L(\theta) - \mathcal L(\theta').$
> Applying the exponential function to both sides yields: $
> \Pr_{\theta'}[S] \exp(\mathcal L (\theta') - \mathcal L(\theta)) \leq \Pr_{\theta}[S].$
> Thus, the result follows.
> You are absolutely correct that $\Pr_{\theta'}[S] > \Pr_{\theta}[S]$ may indeed hold, but only under the condition, as the proof demonstrates, that $\mathcal{L}(\theta') < \mathcal{L}(\theta)$.
>
>
>
>
> -Line 490: “SE(K,β)”->“SE(K2,β)”
>
> $\textbf{Response:}$ We fixed it, thank you.
>
>
>
> I believe the authors should make the assumptions they need to use clearer from the introduction to improve the presentation quality. For example, the strong convexity and smoothness assumption is not mentioned in the statement of Theorem 1.5, which makes it seem too strong.
>
> $\textbf{Response:}$ Thank you for the suggestion, it is indeed something that should have been included in the theorem.
>
> Questions:
> In section 1.1 (line 133), you mention that: “the fact that our algorithm only requires an α-truncated sample oracle is what enables us to achieve a polynomial number of oracle calls to the membership oracle to S”. Can you elaborate on that? Is this not due to the threshold that you introduce in the optimization problems? It seems that assuming the truncated oracle access only allows logarithmic dependence instead of polynomial, which you would get via rejection sampling.
>
> $\textbf{Response:}$ You are correct; this is due to the optimization problems we introduce. These optimization problems ensure that the algorithm operates using only an $\alpha$-truncated oracle, with each call to the oracle assumed to always produce an answer. Please let us know if this does not fully address your concern.

---

### Meta-Review · Area_Chair_ZJhP · 2024-12-18

**Metareview:**

The paper focuses on the problem of parameter estimation from truncated samples, i.e., samples that lie inside a survival set S and moreover it is assumed that we have oracle access to this set S. The main result of this work is to design an algorithm that is polynomial in 1/\alpha where \alpha is the probability mass of S and improve prior works in which the running time was exponential in 1/\alpha.
From a technical standpoint, the novelty lies in the definition of a sequence of constrained optimization problems. Specifically, the authors use projected SGD in a region where the negative log likelihood is not allowed to exceed a certain threshold during each iteration. This prohibits the algorithm from moving to regions in the parameter space that would assign very small mass to the truncation set and seems to be crucial for achieving the exponential improvement in terms of 1/\alpha. The reviewers and so does the AC believe that the paper is above the bar and has nice technical contributions, it improves the state-of-the-art for learning from truncated samples. We recommend the reviewers to read carefully 3xik comments and improve the presentation for the camera ready.

**Additional Comments On Reviewer Discussion:**

The main point raised was the presentation of the proofs and the fact that there are few typos. Nevertheless, all these issues seem addressable for the camera ready version.

---

### Decision · Program_Chairs · 2025-01-22

Accept (Poster)